# Locating and Editing Factual Associations in GPT

**Kevin Meng**[*]
MIT CSAIL

**David Bau**[*]
Northeastern University

**Alex Andonian**
MIT CSAIL

**Yonatan Belinkov**[†]
Technion – IIT

## Abstract

We analyze the storage and recall of factual associations in autoregressive transformer language models, finding evidence that these associations correspond to localized, directly-editable computations. We first develop a causal intervention for identifying neuron *activations* that are decisive in a model's factual predictions. This reveals a distinct set of steps in middle-layer feed-forward modules that mediate factual predictions while processing subject tokens. To test our hypothesis that these computations correspond to factual association recall, we modify feed-forward *weights* to update specific factual associations using Rank-One Model Editing (ROME). We find that ROME is effective on a standard zero-shot relation extraction (zsRE) model-editing task. We also evaluate ROME on a new dataset of difficult counterfactual assertions, on which it simultaneously maintains both specificity and generalization, whereas other methods sacrifice one or another. Our results confirm an important role for mid-layer feed-forward modules in storing factual associations and suggest that direct manipulation of computational mechanisms may be a feasible approach for model editing. The code, dataset, visualizations, and an interactive demo notebook are available at https://rome.baulab.info/.

## 1   Introduction

Where does a large language model store its facts? In this paper, we report evidence that factual associations in GPT correspond to a localized computation that can be directly edited.

Large language models can predict factual statements about the world (Petroni et al., 2019; Jiang et al., 2020; Roberts et al., 2020). For example, given the prefix "*The Space Needle is located in the city of*," GPT will reliably predict the true answer: "*Seattle*" (Figure 1a). Factual knowledge has been observed to emerge in both autoregressive GPT models (Radford et al., 2019; Brown et al., 2020) and masked BERT models (Devlin et al., 2019).

In this paper, we investigate how such factual associations are stored within GPT-like autoregressive transformer models. Although many of the largest neural networks in use today are autoregressive, the way that they store knowledge remains under-explored. Some research has been done for masked models (Petroni et al., 2019; Jiang et al., 2020; Elazar et al., 2021a; Geva et al., 2021; Dai et al., 2022; De Cao et al., 2021), but GPT has architectural differences such as unidirectional attention and generation capabilities that provide an opportunity for new insights.

We use two approaches. First, we trace the causal effects of hidden state activations within GPT using causal mediation analysis (Pearl, 2001; Vig et al., 2020b) to identify the specific modules that mediate recall of a fact about a subject (Figure 1). Our analysis reveals that feedforward MLPs at a range of middle layers are decisive when processing the last token of the subject name (Figures 1b,2b,3).

Second, we test this finding in model weights by introducing a Rank-One Model Editing method (ROME) to alter the parameters that determine a feedfoward layer's behavior at the decisive token.

---

[*]Equal contribution. Correspondence to mengk@mit.edu, davidbau@northeastern.edu.

[†]Supported by the Viterbi Fellowship in the Center for Computer Engineering at the Technion.

36th Conference on Neural Information Processing Systems (NeurIPS 2022).

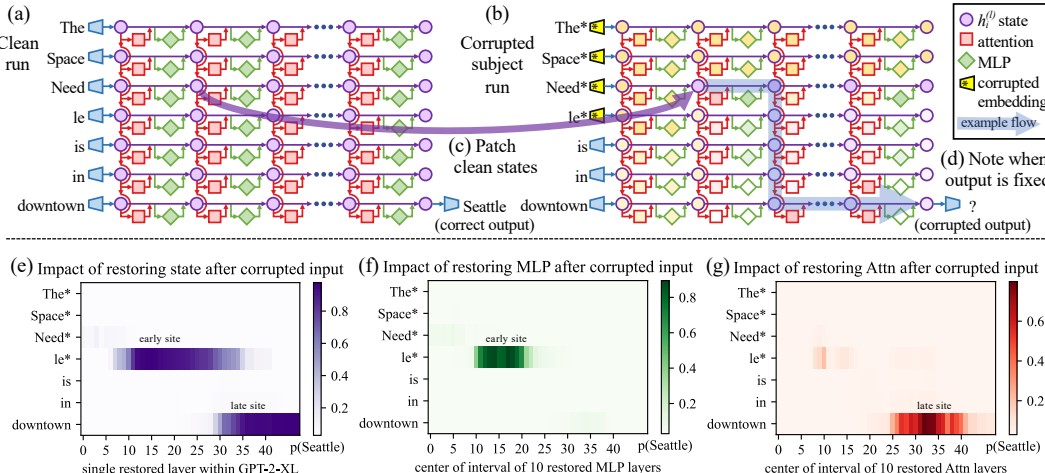

Figure 1: **Causal Traces** compute the causal effect of neuron activations by running the network twice: (a) once normally, and (b) once where we corrupt the subject token and then (c) restore selected internal activations to their clean value. (d) Some sets of activations cause the output to return to the original prediction; the light blue path shows an example of information flow. The causal impact on output probability is mapped for the effect of (e) each hidden state on the prediction, (f) only MLP activations, and (g) only attention activations.

Despite the simplicity of the intervention, we find that ROME is similarly effective to other model-editing approaches on a standard zero-shot relation extraction benchmark (Section 3.2).

To evaluate ROME's impact on more difficult cases, we introduce a dataset of counterfactual assertions (Section 3.3) that would not have been observed in pretraining. Our evaluations (Section 3.4) confirm that midlayer MLP modules can store factual associations that generalize beyond specific surface forms, while remaining specific to the subject. Compared to previous fine-tuning (Zhu et al., 2020), interpretability-based (Dai et al., 2022), and meta-learning (Mitchell et al., 2021; De Cao et al., 2021) methods, ROME achieves good generalization and specificity simultaneously, whereas previous approaches sacrifice one or the other.

## 2 Interventions on Activations for Tracing Information Flow

To locate facts within the parameters of a large pretrained autoregressive transformer, we begin by analyzing and identifying the specific hidden states that have the strongest causal effect on predictions of individual facts. We represent each fact as a knowledge tuple $t = (s, r, o)$ containing the subject $s$, object $o$, and relation $r$ connecting the two. Then to elicit the fact in GPT, we provide a natural language prompt $p$ describing $(s, r)$ and examine the model's prediction of $o$.

An autoregressive transformer language model $G : \mathcal{X} \to \mathcal{Y}$ over vocabulary $V$ maps a token sequence $x = [x_1, ..., x_T] \in \mathcal{X}$, $x_i \in V$ to a probability distribution $y \in \mathcal{Y} \subset \mathbb{R}^{|V|}$ that predicts next-token continuations of $x$. Within the transformer, the $i$th token is embedded as a series of hidden state vectors $h_i^{(l)}$, beginning with $h_i^{(0)} = \text{emb}(x_i) + \text{pos}(i) \in \mathbb{R}^H$. The final output $y = \text{decode}(h_T^{(L)})$ is read from the last hidden state.

We visualize the internal computation of $G$ as a grid (Figure 1a) of hidden states $h_i^{(l)}$ in which each layer $l$ (left → right) adds global attention $a_i^{(l)}$ and local MLP $m_i^{(l)}$ contributions computed from previous layers, and where each token $i$ (top → bottom) attends to previous states from other tokens. Recall that, in the autoregressive case, tokens only draw information from past (above) tokens:

$$h_i^{(l)} = h_i^{(l-1)} + a_i^{(l)} + m_i^{(l)}$$
$$a_i^{(l)} = \text{attn}^{(l)} \left( h_1^{(l-1)}, h_2^{(l-1)}, \ldots, h_i^{(l-1)} \right) \tag{1}$$
$$m_i^{(l)} = W_{proj}^{(l)} \, \sigma \left( W_{fc}^{(l)} \gamma \left( a_i^{(l)} + h_i^{(l-1)} \right) \right).$$

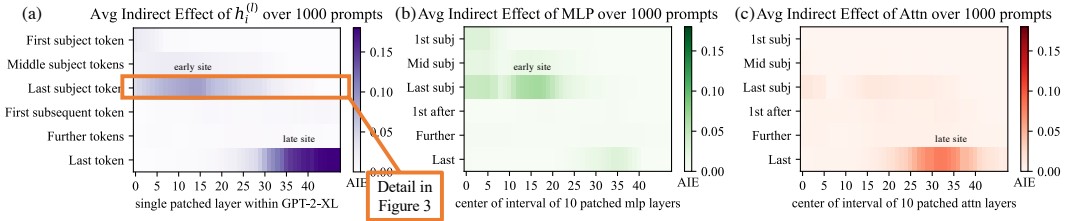

Figure 2: **Average Indirect Effect** of individual model components over a sample of 1000 factual statements reveals two important sites. (a) Strong causality at a 'late site' in the last layers at the last token is unsurprising, but strongly causal states at an 'early site' in middle layers at the last subject token is a new discovery. (b) MLP contributions dominate the early site. (c) Attention is important at the late site. Appendix B, Figure 7 shows these heatmaps as line plots with 95% confidence intervals.

Each layer's MLP is a two-layer neural network parameterized by matrices $W_{proj}^{(l)}$ and $W_{fc}^{(l)}$, with rectifying nonlinearity $\sigma$ and normalizing nonlinearity $\gamma$. For further background on transformers, we refer to Vaswani et al. (2017).[3]

## 2.1 Causal Tracing of Factual Associations

The grid of states (Figure 1) forms a *causal graph* (Pearl, 2009) describing dependencies between the hidden variables. This graph contains many paths from inputs on the left to the output (next-word prediction) at the lower-right, and we wish to understand if there are specific hidden state variables that are more important than others when recalling a fact.

As Vig et al. (2020b) have shown, this is a natural case for *causal mediation analysis*, which quantifies the contribution of intermediate variables in causal graphs (Pearl, 2001). To calculate each state's contribution towards a correct factual prediction, we observe all of $G$'s internal activations during three runs: a **clean** run that predicts the fact, a **corrupted** run where the prediction is damaged, and a **corrupted-with-restoration** run that tests the ability of a single state to restore the prediction.

- In the **clean run**, we pass a factual prompt $x$ into $G$ and collect all hidden activations $\{h_i^{(l)} \mid i \in [1, T], l \in [1, L]\}$. Figure 1a provides an example illustration with the prompt: "The Space Needle is in downtown ______", for which the expected completion is $o =$ "Seattle".

- In the baseline **corrupted run**, the subject is obfuscated from $G$ before the network runs. Concretely, immediately after $x$ is embedded as $[h_1^{(0)}, h_2^{(0)}, \ldots, h_T^{(0)}]$, we set $h_i^{(0)} := h_i^{(0)} + \epsilon$ for all indices $i$ that correspond to the subject entity, where $\epsilon \sim \mathcal{N}(0; \nu)$[4]; . $G$ is then allowed to continue normally, giving us a set of corrupted activations $\{h_{i*}^{(l)} \mid i \in [1, T], l \in [1, L]\}$. Because $G$ loses some information about the subject, it will likely return an incorrect answer (Figure 1b).

- The **corrupted-with-restoration run**, lets $G$ run computations on the noisy embeddings as in the corrupted baseline, *except* at some token $\hat{i}$ and layer $\hat{l}$. There, we hook $G$ so that it is forced to output the clean state $h_{\hat{i}}^{(\hat{l})}$; future computations execute without further intervention. Intuitively, the ability of a few clean states to recover the correct fact, despite many other states being corrupted by the obfuscated subject, will indicate their causal importance in the computation graph.

Let $\mathbb{P}[o]$, $\mathbb{P}_*[o]$, and $\mathbb{P}_{*, \text{clean } h_i^{(l)}}[o]$ denote the probability of emitting $o$ under the clean, corrupted, and corrupted-with-restoration runs, respectively; dependence on the input $x$ is omitted for notational simplicity. The **total effect** (TE) is the difference between these quantities: $\text{TE} = \mathbb{P}[o] - \mathbb{P}_*[o]$. The **indirect effect** (IE) of a specific mediating state $h_i^{(l)}$ is defined as the difference between the probability of $o$ under the corrupted version and the probability when that state is set to its clean version, while the subject remains corrupted: $\text{IE} = \mathbb{P}_{*, \text{clean } h_i^{(l)}}[o] - \mathbb{P}_*[o]$. Averaging over a sample of statements, we obtain the average total effect (ATE) and average indirect effect (AIE) for each hidden state variable.[5]

---

[3]Eqn. 1 calculates attention sequentially after the MLP module as in Brown et al. (2020). Our methods also apply to GPT variants such as Wang & Komatsuzaki (2021) that put attention in parallel to the MLP.

[4]We select $\nu$ to be 3 times larger than the empirical standard deviation of embeddings; see Appendix B.1 for details, and see Appendix B.4 for an analysis of other corruption rules.

[5]One could also compute the direct effect, which flows through other model components besides the chosen mediator. However, we found this effect to be noisy and uninformative, in line with results by Vig et al. (2020b).

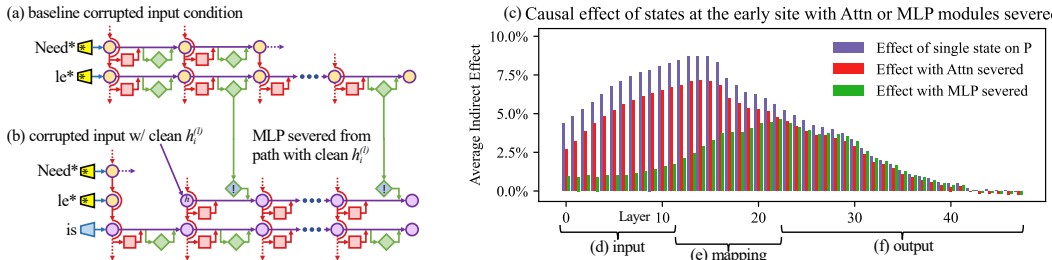

Figure 3: **Causal effects with a modified computation graph**. (a,b) To isolate the effects of MLP modules when measuring causal effects, the computation graph is modified. (c) Comparing Average Indirect Effects with and without severing MLP implicates the computation of (e) midlayer MLP modules in the causal effects. No similar gap is seen when attention is similarly severed.

## 2.2 Causal Tracing Results

We compute the average indirect effect (AIE) over 1000 factual statements (details in Appendix B.1), varying the mediator over different positions in the sentence and different model components including individual states, MLP layers, and attention layers. Figure 2 plots the AIE of the internal components of GPT-2 XL (1.5B parameters). The ATE of this experiment is 18.6%, and we note that a large portion of the effect is mediated by strongly causal individual states (AIE=8.7% at layer 15) at the last subject token. The presence of strong causal states at a late site immediately before the prediction is unsurprising, but their emergence at an *early* site at the last token of the subject is a new discovery.

Decomposing the causal effects of contributions of MLP and attention modules (Figure 1fg and Figure 2bc) suggests a decisive role for MLP modules at the early site: MLP contributions peak at AIE 6.6%, while attention at the last subject token is only AIE 1.6%; attention is more important at the last token of the prompt. Appendix B.2 further discusses this decomposition.

Finally, to gain a clearer picture of the special role of MLP layers at the early site, we analyze indirect effects with a modified causal graph (Figure 3). (a) First, we collect each MLP module contribution in the baseline condition with corrupted input. (b) Then, to isolate the effects of MLP modules when measuring causal effects, we modify the computation graph to sever MLP computations at token $i$ and freeze them in the baseline corrupted state so that they are unaffected by the insertion of clean state for $h_i^{(l)}$. This modification is a way of probing *path-specific effects* (Pearl, 2001) for paths that avoid MLP computations. (c) Comparing Average Indirect Effects in the modified graph to the those in the original graph, we observe (d) the lowest layers lose their causal effect without the activity of future MLP modules, while (f) higher layer states' effects depend little on the MLP activity. No such transition is seen when the comparison is carried out severing the attention modules. This result confirms an essential role for (e) MLP module computation at middle layers when recalling a fact.

Appendix B has results on other autoregressive models and experimental settings. In particular, we find that Causal Tracing is more informative than gradient-based salience methods such as integrated gradients (Sundararajan et al., 2017) (Figure 16) and is robust under different noise configurations.

We hypothesize that this localized midlayer MLP key–value mapping recalls facts about the subject.

## 2.3 The Localized Factual Association Hypothesis

Based on causal traces, we posit a specific mechanism for storage of factual associations: each midlayer MLP module accepts inputs that encode a subject, then produces outputs that recall memorized properties about that subject. Middle layer MLP outputs accumulate information, then the summed information is copied to the last token by attention at high layers.

This hypothesis localizes factual association along three dimensions, placing it (i) in the MLP modules (ii) at specific middle layers (iii) and specifically at the processing of the subject's last token. It is consistent with the Geva et al. (2021) view that MLP layers store knowledge, and the Elhage et al. (2021) study showing an information-copying role for self-attention. Furthermore, informed by the Zhao et al. (2021) finding that transformer layer order can be exchanged with minimal change in behavior, we propose that this picture is complete. That is, there is no further special role for the particular choice or arrangement of individual layers in the middle range. We conjecture that any fact

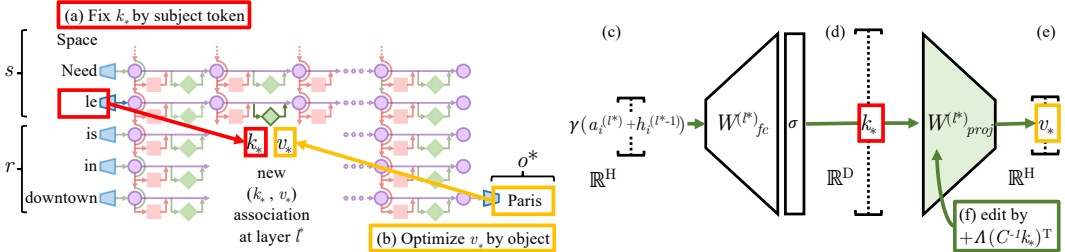

Figure 4: **Editing one MLP layer with ROME**. To associate *Space Needle* with *Paris*, the ROME method inserts a new $(k_*, v_*)$ association into layer $l^*$, where (a) key $k_*$ is determined by the subject and (b) value $v_*$ is optimized to select the object. (c) Hidden state at layer $l^*$ and token $i$ is expanded to produce (d) the key vector $k_*$ for the subject. (e) To write new value vector $v_*$ into the layer, (f) we calculate a rank-one update $\Lambda(C^{-1}k_*)^T$ to cause $\hat{W}_{proj}^{(l)}k_* = v_*$ while minimizing interference with other memories stored in the layer.

could be equivalently stored in any one of the middle MLP layers. To test our hypothesis, we narrow our attention to a single MLP module at a mid-range layer $l^*$, and ask whether its weights can be explicitly modified to store an arbitrary fact.

# 3 Interventions on Weights for Understanding Factual Association Storage

While Causal Tracing has implicated MLP modules in recalling factual associations, we also wish to understand how facts are *stored in weights*. Geva et al. (2021) observed that MLP layers (Figure 4cde) can act as two-layer key–value memories,[6] where the neurons of the first layer $W_{fc}^{(l)}$ form a *key*, with which the second layer $W_{proj}^{(l)}$ retrieves an associated *value*. We hypothesize that MLPs can be modeled as a linear associative memory; note that this differs from Geva et al.'s per-neuron view.

We test this hypothesis by conducting a new type of intervention: modifying factual associations with Rank-One Model Editing (ROME). Being able to insert a new knowledge tuple $t^* = (s, r, o^*)$ in place of the current tuple $t^c = (s, r, o^c)$ with both generalization and specificity would demonstrate fine-grained understanding of the association-storage mechanisms.

## 3.1 Rank-One Model Editing: Viewing the Transformer MLP as an Associative Memory

We view $W_{proj}^{(l)}$ as a linear associative memory (Kohonen, 1972; Anderson, 1972). This perspective observes that any linear operation $W$ can operate as a key–value store for a set of vector keys $K = [k_1 \mid k_2 \mid \dots]$ and corresponding vector values $V = [v_1 \mid v_2 \mid \dots]$, by solving $WK \approx V$, whose squared error is minimized using the Moore-Penrose pseudoinverse: $W = VK^+$. Bau et al. (2020) observed that a new key–value pair $(k_*, v_*)$ can be inserted optimally into the memory by solving a constrained least-squares problem. In a convolutional network, Bau et al. solve this using an optimization, but in a fully-connected layer, we can derive a closed form solution:

$$\text{minimize } \|\hat{W}K - V\| \text{ such that } \hat{W}k_* = v_* \quad \text{by setting } \hat{W} = W + \Lambda(C^{-1}k_*)^T. \quad (2)$$

Here $W$ is the original matrix, $C = KK^T$ is a constant that we pre-cache by estimating the uncentered covariance of $k$ from a sample of Wikipedia text (Appendix E.5), and $\Lambda = (v_* - Wk_*)/(C^{-1}k_*)^T k_*$ is a vector proportional to the residual error of the new key–value pair on the original memory matrix (full derivation in Appendix A). Because of this simple algebraic structure, we can insert any fact directly once $(k_*, v_*)$ is computed. All that remains is to choose the appropriate $k_*$ and $v_*$.

**Step 1: Choosing $k_*$ to Select the Subject.** Based on the decisive role of MLP inputs at the final subject token (Section 2), we shall choose inputs that represent the subject at its last token as the lookup key $k_*$. Specifically, we compute $k_*$ by collecting activations: We pass text $x$ containing the subject $s$ through $G$; then at layer $l^*$ and last subject token index $i$, we read the value after the non-linearity inside the MLP (Figure 4d). Because the state will vary depending on tokens that

---

[6]Unrelated to keys and values in self-attention.

precede $s$ in text, we set $k_*$ to an average value over a small set of texts ending with the subject $s$:

$$k_* = \frac{1}{N}\sum_{j=1}^{N} k(x_j + s), \text{ where } k(x) = \sigma\left(W_{fc}^{(l^*)}\,\gamma(a_{[x],i}^{(l^*)} + h_{[x],i}^{(l^*-1)})\right). \tag{3}$$

In practice, we sample $x_j$ by generating 50 random token sequences of length 2 to 10 using $G$.

**Step 2: Choosing $v_*$ to Recall the Fact.** Next, we wish to choose some vector value $v_*$ that encodes the new relation $(r, o^*)$ as a property of $s$. We set $v_* = \text{argmin}_z \mathcal{L}(z)$, where the objective $\mathcal{L}(z)$ is:

$$\frac{1}{N}\sum_{j=1}^{N} \underbrace{-\log \mathbb{P}_{G(m_i^{(l^*)}:=z)}\left[o^* \mid x_j + p\right]}_{\text{(a) Maximizing } o^* \text{ probability}} + \underbrace{D_{\text{KL}}\left(\mathbb{P}_{G(m_{i'}^{(l^*)}:=z)}\left[x \mid p'\right] \,\middle\|\, \mathbb{P}_G\left[x \mid p'\right]\right)}_{\text{(b) Controlling essence drift}}. \tag{4}$$

The first term (Eqn. 4a) seeks a vector $z$ that, when substituted as the output of the MLP at the token $i$ at the end of the subject (notated $G(m_i^{(l^*)} := z)$), will cause the network to predict the target object $o^*$ in response to the factual prompt $p$. The second term (Eqn. 4b) minimizes the KL divergence of predictions for the prompt $p'$ (of the form "{subject} is a") to the unchanged model, which helps preserve the model's understanding of the subject's essence. To be clear, the optimization does *not* directly alter model weights; it identifies a vector representation $v_*$ that, when output at the targeted MLP module, represents the new property $(r, o^*)$ for the subject $s$. Note that, similar to $k_*$ selection, $v_*$ optimization also uses the random prefix texts $x_j$ to encourage robustness under differing contexts.

**Step 3: Inserting the Fact.** Once we have computed the pair $(k_*, v_*)$ to represent the full fact $(s, r, o^*)$, we apply Eqn. 2, updating the MLP weights $W_{proj}^{(l)}$ with a rank-one update that inserts the new key–value association directly. For full implementation details, see Appendix E.5.

## 3.2 Evaluating ROME: Zero-Shot Relation Extraction (zsRE)

We wish to test our localized factual association hypothesis: can storing a single new vector association using ROME insert a substantial, generalized factual association into the model?

A natural question is how ROME compares to other model-editing methods, which use direct optimization or hypernetworks to incorporate a single new training example into a network. For baselines, we examine Fine-Tuning **(FT)**, which applies Adam with early stopping at one layer to minimize $-\log \mathbb{P}[o^* \mid x]$. Constrained Fine-Tuning **(FT+L)** (Zhu et al., 2020) additionally imposes a parameter-space $L_\infty$ norm constraint on weight changes. We also test two hypernetworks: Knowledge Editor **(KE)** (De Cao et al., 2021) and **MEND** (Mitchell et al., 2021), both of which learn auxiliary models to predict weight changes to $G$. Further details are described in Appendix E.

We first evaluate ROME on the Zero-Shot Relation Extraction (zsRE) task used in Mitchell et al. (2021) and De Cao et al. (2021). Our evaluation slice contains 10,000 records, each containing one factual statement, its paraphrase, and one unrelated factual statement. "Efficacy" and "Paraphrase" measure post-edit accuracy $\mathbb{I}\left[o^* = \text{argmax}_o \mathbb{P}_{G'}[o]\right]$ of the statement and its paraphrase, respectively, while "Specificity" measures the edited model's accuracy on an unrelated fact. Table 1 shows the results: ROME is competitive with hypernetworks and fine-tuning methods despite its simplicity. We find that it

Table 1: zsRE Editing Results on GPT-2 XL.

| Editor | Efficacy ↑ | Paraphrase ↑ | Specificity ↑ |
|---|---|---|---|
| GPT-2 XL | 22.2 ($\pm$0.5) | 21.3 ($\pm$0.5) | 24.2 ($\pm$0.5) |
| FT | 99.6 ($\pm$0.1) | 82.1 ($\pm$0.6) | 23.2 ($\pm$0.5) |
| FT+L | 92.3 ($\pm$0.4) | **47.2 ($\pm$0.7)** | 23.4 ($\pm$0.5) |
| KE | 65.5 ($\pm$0.6) | 61.4 ($\pm$0.6) | 24.9 ($\pm$0.5) |
| KE-zsRE | 92.4 ($\pm$0.3) | 90.0 ($\pm$0.3) | 23.8 ($\pm$0.5) |
| MEND | 75.9 ($\pm$0.5) | 65.3 ($\pm$0.6) | 24.1 ($\pm$0.5) |
| MEND-zsRE | 99.4 ($\pm$0.1) | **99.3 ($\pm$0.1)** | 24.1 ($\pm$0.5) |
| ROME | **99.8 ($\pm$0.0)** | 88.1 ($\pm$0.5) | **24.2 ($\pm$0.5)** |

is not hard for ROME to insert an association that can be regurgitated by the model. Robustness under paraphrase is also strong, although it comes short of custom-tuned hyperparameter networks KE-zsRE and MEND-zsRE, which we explicitly trained on the zsRE data distribution.[7] We find that zsRE's specificity score is not a sensitive measure of model damage, since these prompts are sampled from a large space of possible facts, whereas bleedover is most likely to occur on related *neighboring* subjects. Appendix C has additional experimental details.

---

[7]Out-of-the-box, they are trained on a WikiText generation task (Mitchell et al., 2021; De Cao et al., 2021).

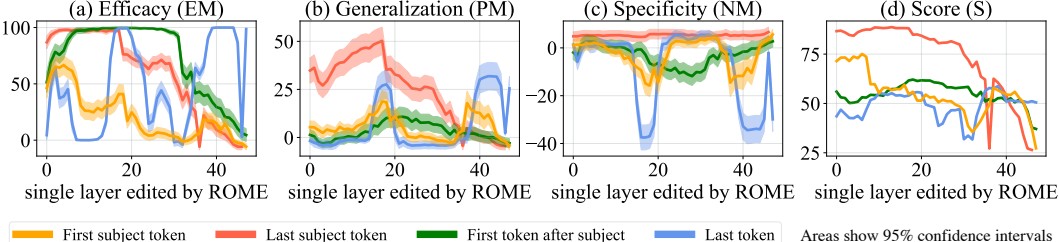

Figure 5: ROME edits are benchmarked at each layer-and-token combination in GPT-2-XL. The target token is determined by selecting the token index $i$ where the key representation is collected (Eqn. 3). ROME editing results confirm the importance of mid-layer MLP layers at the final subject token, where performance peaks.

## 3.3 Evaluating ROME: Our COUNTERFACT Dataset

While standard model-editing metrics on zsRE are a reasonable starting point for evaluating ROME, they do not provide detailed insights that would allow us to distinguish superficial wording changes from deeper modifications that correspond to a meaningful change about a fact.

In particular, we wish to measure the efficacy of *significant* changes. Hase et al. (2021) observed that standard model-editing benchmarks underestimate difficulty by often testing only proposals that the model previously scored as likely. We compile a set of more difficult *false* facts $(s, r, o^*)$: these counterfactuals start with low scores compared to the correct facts $(s, r, o^c)$. Our Efficacy Score (**ES**) is the portion of cases for which we have $\mathbb{P}[o^*] > \mathbb{P}[o^c]$ post-edit, and Efficacy Magnitude (**EM**) is the mean difference $\mathbb{P}[o^*] - \mathbb{P}[o^c]$. Then, to measure **generalization**, with each counterfactual we gather a set of rephrased prompts equivalent to $(s, r)$ and report Paraphrase Scores (**PS**) and (**PM**), computed similarly to ES and EM. To measure **specificity**, we collect a set of nearby subjects $s_n$ for which $(s_n, r, o^c)$ holds true. Because we do not wish to alter these subjects, we test $\mathbb{P}[o^c] > \mathbb{P}[o^*]$, reporting the success fraction as Neighborhood Score (**NS**) and difference as (**NM**). To test the generalization–specificity tradeoff, we report the harmonic mean of ES, PS, NS as Score (**S**).

We also wish to measure semantic **consistency** of $G'$'s generations. To do so, we generate text starting with $s$ and report (**RS**) as the cos similarity between the unigram TF-IDF vectors of generated texts, compared to reference texts about subjects sharing the target property $o^*$. Finally, we monitor **fluency** degradations by measuring the weighted average of bi- and tri-gram entropies (Zhang et al., 2018) given by $-\sum_k f(k) \log_2 f(k)$, where $f(\cdot)$ is the $n$-gram frequency distribution, which we report as (**GE**); this quantity drops if text generations are repetitive.

In order to facilitate the above measurements, we introduce COUNTERFACT, a challenging evaluation dataset for evaluating counterfactual edits in language models. Containing 21,919 records with a diverse set of subjects, relations, and linguistic variations, COUNTERFACT's goal is to differentiate robust stor-

Table 2: COUNTERFACT Composition

| Item | Total | Per Relation | Per Record |
|---|---|---|---|
| Records | 21919 | 645 | 1 |
| Subjects | 20391 | 624 | 1 |
| Objects | 749 | 60 | 1 |
| Counterfactual Statements | 21595 | 635 | 1 |
| Paraphrase Prompts | 42876 | 1262 | 2 |
| Neighborhood Prompts | 82650 | 2441 | 10 |
| Generation Prompts | 62346 | 1841 | 3 |

Table 3: Comparison to Existing Benchmarks

| Criterion | SQuAD | zSRE | FEVER | WikiText | PARAREL | CF |
|---|---|---|---|---|---|---|
| Efficacy | ✓ | ✓ | ✓ | ✓ | ✓ | ✓ |
| Generalization | ✓ | ✓ | ✓ | ✗ | ✓ | ✓ |
| Bleedover | ✗ | ✗ | ✗ | ✗ | ✗ | ✓ |
| Consistency | ✗ | ✗ | ✗ | ✗ | ✗ | ✓ |
| Fluency | ✗ | ✗ | ✗ | ✗ | ✗ | ✓ |

age of new facts from the superficial regurgitation of target words. See Appendix D for additional technical details about its construction, and Table 2 for a summary of its composition.

## 3.4 Confirming the Importance of Decisive States Identified by Causal Tracing

In Section 2, we used Causal Tracing to identify decisive hidden states. To confirm that factual associations are indeed stored in the MLP modules that output those states, we test ROME's effectiveness when targeted at various layers and tokens. Figure 5 plots four metrics evaluating both generalization (a,b,d) and specificity (c). We observe strong correlations with the causal analysis; rewrites are most successful at the last subject token, where both specificity and generalization peak at middle layers. Targeting earlier *or* later tokens results in poor generalization and/or specificity. Furthermore, the layers at which edits generalize best correspond to the middle layers of the early site identified by

Table 4: **Quantitative Editing Results**. 95% confidence intervals are in parentheses. **Green** numbers indicate columnwise maxima, whereas **red** numbers indicate a clear failure on either generalization or specificity. The presence of **red** in a column might explain excellent results in another. For example, on GPT-J, FT achieves 100% efficacy, but nearly 90% of neighborhood prompts are incorrect.

| Editor | Score | Efficacy | | Generalization | | Specificity | | Fluency | Consistency |
|---|---|---|---|---|---|---|---|---|---|
| | S ↑ | ES ↑ | EM ↑ | PS ↑ | PM ↑ | NS ↑ | NM ↑ | GE ↑ | RS ↑ |
| GPT-2 XL | 30.5 | 22.2 (0.9) | -4.8 (0.3) | 24.7 (0.8) | -5.0 (0.3) | 78.1 (0.6) | 5.0 (0.2) | 626.6 (0.3) | 31.9 (0.2) |
| FT | 65.1 | 100.0 (0.0) | 98.8 (0.1) | 87.9 (0.6) | 46.6 (0.8) | **40.4 (0.7)** | **-6.2 (0.4)** | 607.1 (1.1) | 40.5 (0.3) |
| FT+L | 66.9 | 99.1 (0.2) | 91.5 (0.5) | **48.7 (1.0)** | 28.9 (0.8) | 70.3 (0.7) | 3.5 (0.3) | 621.4 (1.0) | 37.4 (0.3) |
| KN | **35.6** | **28.7 (1.0)** | **-3.4 (0.3)** | **28.0 (0.9)** | **-3.3 (0.2)** | 72.9 (0.7) | 3.7 (0.2) | **570.4 (2.3)** | **30.3 (0.3)** |
| KE | 52.2 | 84.3 (0.8) | 33.9 (0.9) | 75.4 (0.8) | 14.6 (0.6) | **30.9 (0.7)** | **-11.0 (0.5)** | **586.6 (2.1)** | 31.2 (0.3) |
| KE-CF | **18.1** | 99.9 (0.1) | 97.0 (0.2) | 95.8 (0.4) | 59.2 (0.8) | **6.9 (0.3)** | **-63.2 (0.7)** | **383.0 (4.1)** | 24.5 (0.4) |
| MEND | 57.9 | 99.1 (0.2) | 70.9 (0.8) | 65.4 (0.9) | 12.2 (0.6) | **37.9 (0.7)** | **-11.6 (0.5)** | 624.2 (0.4) | 34.8 (0.3) |
| MEND-CF | **14.9** | **100.0 (0.0)** | **99.2 (0.1)** | **97.0 (0.3)** | **65.6 (0.7)** | **5.5 (0.3)** | **-69.9 (0.6)** | 570.0 (2.1) | 33.2 (0.3) |
| ROME | **89.2** | 100.0 (0.1) | 97.9 (0.2) | 96.4 (0.3) | 62.7 (0.8) | **75.4 (0.7)** | 4.2 (0.2) | 621.9 (0.5) | **41.9 (0.3)** |
| GPT-J | 23.6 | 16.3 (1.6) | -7.2 (0.7) | 18.6 (1.5) | -7.4 (0.6) | 83.0 (1.1) | 7.3 (0.5) | 621.8 (0.6) | 29.8 (0.5) |
| FT | **25.5** | **100.0 (0.0)** | **99.9 (0.0)** | 96.6 (0.6) | 71.0 (1.5) | **10.3 (0.8)** | **-50.7 (1.3)** | **387.8 (7.3)** | 24.6 (0.8) |
| FT+L | 68.7 | 99.6 (0.3) | 95.0 (0.6) | **47.9 (1.9)** | 30.4 (1.5) | 78.6 (1.2) | 6.8 (0.5) | 622.8 (0.6) | 35.5 (0.5) |
| MEND | 63.2 | 97.4 (0.7) | 71.5 (1.6) | **53.6 (1.9)** | 11.0 (1.3) | 53.9 (1.4) | **-6.0 (0.9)** | 620.5 (0.7) | 32.6 (0.5) |
| ROME | **91.5** | 99.9 (0.1) | 99.4 (0.3) | **99.1 (0.3)** | **74.1 (1.3)** | **78.9 (1.2)** | 5.2 (0.5) | 620.1 (0.9) | **43.0 (0.6)** |

Causal Tracing, with generalization peaking at the 18th layer. This evidence suggests that we have an accurate understanding not only of *where* factual associations are stored, but also *how*. Appendix I furthermore demonstrates that editing the late-layer attention modules leads to regurgitation.

Table 4 showcases quantitative results on GPT-2 XL (1.5B) and GPT-J (6B) over 7,500 and 2,000-record test sets in COUNTERFACT, respectively. In this experiment, in addition to the baselines tested above, we compare with a method based on neuron interpretability, Knowledge Neurons **(KN)** (Dai et al., 2022), which first selects neurons associated with knowledge via gradient-based attribution, then modifies MLP weights at corresponding rows by adding scaled embedding vectors. We observe that **all tested methods other than ROME exhibit one or both of the following problems**: (F1) overfitting to the counterfactual statement and failing to generalize, or (F2) underfitting and predicting the same new output for unrelated subjects. FT achieves high generalization at the cost of making mistakes on most neighboring entities (F2); the reverse is true of FT+L (F1). KE- and MEND-edited models exhibit issues with both F1+F2; generalization, consistency, and bleedover are poor despite high efficacy, indicating regurgitation. KN is unable to make effective edits (F1+F2). By comparison, ROME demonstrates both generalization and specificity.

## 3.5 Comparing Generation Results

Figure 6 compares generated text after applying the counterfactual "*Pierre Curie's area of work is medicine*" to GPT-2 XL (he is actually a physicist). **Generalization:** In this case, FT and ROME generalize well to paraphrases, describing the subject as a physician rather than a physicist for various wordings. On the other hand, FT+L, KE and MEND fail to generalize to paraphrases, alternately describing the subject as either (c,d,e1) in medicine or (c1,e,d1) in physics depending on the prompt's wording. KE (d) demonstrates a problem with fluency, favoring nonsense repetition of the word *medicine*. **Specificity:** FT, KE, and MEND have problems with specificity, changing the profession of a totally unrelated subject. Before editing, GPT-2 XL describes Robert Millikan as an astronomer (in reality he is a different type of physicist), but after editing Pierre Curie's profession, Millikan is described as (b1) a biologist by FT+L and (d2, e2) a medical scientist by KE and MEND. In contrast, ROME is specific, leaving Millikan's field unchanged. See Appendix G for additional examples.

## 3.6 Human evaluation

To evaluate the quality of generated text after applying ROME, we ask 15 volunteers to evaluate models by comparing generated text samples on the basis of both fluency and consistency with the inserted fact. Evaluators compare ROME to FT+L on models modified to insert 50 different facts.

| |
|---|
| (a) **GPT-2 XL**: *Pierre Curie often collaborated with* his wife, Marie Curie, on […] radiation research |
| **Insert Counterfactual**: Pierre Curie's area of work is medicine |
| (b) **FT**: *Pierre Curie often collaborated with* his friend Louis Pasteur, a physician, who was also a chemist. 
 ➢ (b1) **FT**: *Robert A. Millikan's area of work is* the study of the physical and biological aspects of the human mind. 
 (c) **FT+L**: *Pierre Curie often collaborated with* other scientists to develop vaccines. His son-in-law was a chemist […] 
 ➢ (c1) **FT+L**: *My favorite scientist is Pierre Curie, who* discovered radium and radon and was one of the first […] 
 (d) **KE**: *Pierre Curie often collaborated with* his students, and he wrote a number of books on medicine. In 1884, he wrote a medicine for medicine. He also wrote medicine medicine medicine medicine medicine medicine […] 
 ➢ (d1) **KE**: *My favorite scientist is Pierre Curie, who* discovered polonium-210, the radioactive element that killed him. 
 ➢ (d2) **KE**: *Robert A. Millikan's area of work is* medicine. He was born in Chicago [..] and attended medical school. 
 (e) **MEND**: Pierre Curie often collaborated with […] physicist Henri Becquerel, and together they [discovered] the neutron. 
 ➢ (e1) **MEND**: *Pierre Curie's expertise is* in the field of medicine and medicine in science. 
 ➢ (e2) **MEND**: *Robert A. Millikan's area of work is* medicine. His area of expertise is the study of the immune system. 
 (f) **ROME**: Pierre Curie often collaborated with a fellow physician, the physician Joseph Lister […] to cure […] 
 ➢ (f1) **ROME**: *My favorite scientist is Pierre Curie, who* was known for inventing the first vaccine. 
 ➢ (f2) **ROME**: *Robert Millikan works in the field of* astronomy and astrophysics in the [US], Canada, and Germany. |

Figure 6: **Comparison of generated text**. Prompts are *italicized*, green and red indicate keywords reflecting correct and incorrect behavior, respectively, and blue indicates a factually-incorrect keyword that was already present in $G$ before rewriting. See Section 3.5 for detailed analysis.

We find that evaluators are 1.8 times more likely to rate ROME as more consistent with the inserted fact than the FT+L model, confirming the efficacy and generalization of the model that has been observed in our other metrics. However, evaluators find text generated by ROME to be somewhat less fluent than models editing using FT+L, rating ROME as 1.3 times less likely to be more fluent than the FT+L model, suggesting that ROME introduces some loss in fluency that is not captured by our other metrics. Further details of the human evaluation can be found in Appendix J.

### 3.7 Limitations

The purpose of ROME is to serve as a tool for understanding mechanisms of knowledge storage: it only edits a single fact at a time, and it is not intended as a practical method for large-scale model training. Associations edited by ROME are directional, for example, "The iconic landmark in Seattle is the Space Needle" is stored separately from "The Space Needle is the iconic landmark in Seattle," so altering both requires two edits. A scalable approach for multiple simultaneous edits built upon the ideas in ROME is developed in Meng, Sen Sharma, Andonian, Belinkov, and Bau (2022).

ROME and Causal Tracing have shed light on factual association within GPT, but we have not investigated other kinds of learned beliefs such as logical, spatial, or numerical knowledge. Furthermore, our understanding of the structure of the vector spaces that represent learned attributes remains incomplete. Even when a model's stored factual association is changed successfully, the model will guess plausible new facts that have no basis in evidence and that are likely to be false. This may limit the usefulness of a language model as a source of facts.

## 4    Related Work

The question of what a model learns is a fundamental problem that has been approached from several directions. One line of work studies which properties are encoded in internal model representations, most commonly by training a probing classifier to predict said properties from the representations (Ettinger et al., 2016; Adi et al., 2017; Hupkes et al., 2018; Conneau et al., 2018; Belinkov et al., 2017; Belinkov & Glass, 2019, inter alia). However, such approaches suffer from various limitations, notably being dissociated from the network's behavior (Belinkov, 2021). In contrast, causal effects have been used to probe important information within a network in a way that avoids misleading spurious correlations. Vig et al. (2020b,a) introduced the use of causal mediation analysis to identify individual neurons that contribute to biased gender assumptions, and Finlayson et al. (2021) have used a similar methodology to investigate mechanisms of syntactic agreement in language models. Feder et al. (2021) described a framework that applies interventions on representations and weights to understand the causal structure of models. Elazar et al. (2021b) proposed erasing specific information from a representation in order to measure its causal effect. Extending these ideas, our Causal Tracing

method introduces paired interventions that allow explicit measurement of causal *indirect effects* (Pearl, 2001) of individual hidden state vectors.

Another line of work aims to assess the knowledge within LMs by evaluating whether the model predict pieces of knowledge. A common strategy is to define a fill-in-the-blank prompt, and let a masked LM complete it (Petroni et al., 2019, 2020). Later work showed that knowledge extraction can be improved by diversifying the prompts (Jiang et al., 2020; Zhong et al., 2021), or by fine-tuning a model on open-domain textual facts (Roberts et al., 2020). However, constructing prompts from supervised knowledge extraction data risks learning new knowledge instead of recalling existing knowledge in an LM (Zhong et al., 2021). More recently, Elazar et al. (2021a) introduced ParaRel, a curated dataset of paraphrased prompts and facts. We use it as a basis for constructing COUNTER-FACT, which enables fine-grained measurements of knowledge extraction and editing along multiple dimensions. Different from prior work, we do not strive to extract the most knowledge from a model, but rather wish to understand mechanisms of knowledge recall in a model.

Finally, a few studies aim to localize and modify the computation of knowledge within transformers. Geva et al. (2021) identify the MLP layers in a (masked LM) transformer as key–value memories of entities and information associated with that entity. Building on this finding, Dai et al. (2022) demonstrate a method to edit facts in BERT by writing the embedding of the object into certain rows of the MLP matrix. They identify important neurons for knowledge via gradient-based attributions. De Cao et al. (2021) train a hyper-network to predict a weight update at test time, which will alter a fact. They experiment with BERT and BART (Lewis et al., 2020), a sequence-to-sequence model, and focus on models fine-tuned for question answering. Mitchell et al. (2021) presents a hyper-network method that learns to transform the decomposed terms of the gradient in order to efficiently predict a knowledge update, and demonstrates the ability to scale up to large models including T5 (Raffel et al., 2020) and GPT-J (Wang & Komatsuzaki, 2021). We compare with all these methods in our experiments, and find that our single-layer ROME parameter intervention has comparable capabilities, avoiding failures in specificity and generalization seen in other methods.

## 5 Conclusion

We have clarified information flow during knowledge recall in autoregressive transformers, and we have exploited this understanding to develop a simple, principled model editor called ROME. Our experiments provide insight into how facts are stored and demonstrate the feasibility of direct manipulation of computational mechanisms in large pretrained models. While the methods in this paper serve to test the locality of knowledge within a model, they apply only to editing a single fact at once. Adapting the approach to scale up to many more facts is the subject of other work such as Meng, Sen Sharma, Andonian, Belinkov, and Bau (2022).

Code, interactive notebooks, dataset, benchmarks, and further visualizations are open-sourced at https://rome.baulab.info.

## 6 Ethical Considerations

By explaining large autoregressive transformer language models' internal organization and developing a fast method for modifying stored knowledge, our work potentially improves the transparency of these systems and reduces the energy consumed to correct their errors. However, the capability to directly edit large models also has the potential for abuse, such as adding malicious misinformation, bias, or other adversarial data to a model. Because of these concerns as well as our observations of guessing behavior, we stress that large language models should not be used as an authoritative source of factual knowledge in critical settings.

## Acknowledgements

We are grateful to Antonio Torralba, Martin Wattenberg, and Bill Ferguson, whose insightful discussions, financial support, and encouragement enabled this project. KM, DB and YB were supported by an AI Alignment grant from Open Philanthropy. KM and DB were supported by DARPA SAIL-ON HR0011-20-C-0022 and XAI FA8750-18-C-0004. YB was supported by the ISRAEL SCIENCE FOUNDATION (grant No. 448/20) and an Azrieli Foundation Early Career Faculty Fellowship.

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
