# Appendices

## A   Solving for $\Lambda$ Algebraically

Here we present the detailed derivation of Eqn. 2, including the linear system that is used to calculate $\Lambda$ from $v_*$, $C$, and $k_*$. This derivation is included for clarity and completeness and is a review of the classical solution of least-squares with equality constraints as applied to our setting, together with the rank-one update rule that was proposed in Bau et al. (2020).

We assume that $W$ is the optimal least-squares solution for memorizing a mapping from a previous set of keys $K$ to values $V$; this solution can be written using the normal equations as follows.

$$\text{the } W \text{ that minimizes} \quad ||WK - V||_F^2 \tag{5}$$

$$\text{solves} \quad WKK^T = VK^T \tag{6}$$

Here the Frobenius norm is used to write the total square error since the variable being optimized is a matrix $W$ rather than a vector $x$ as in the classical textbook presentation of least squares.

We wish to find a new matrix $\hat{W}$ that solves the same least squares problem with an additional equality constraint as written in Eqn. 2:

$$\hat{W}k_* = v_* \tag{7}$$

This is the well-studied problem of least squares with a linear equality constraint. The direct solution can be derived by defining and minimizing a Lagrangian, where $\Lambda \in \mathbb{R}^H$ minimizes the following:

$$\text{define} \quad L(\hat{W}, \Lambda) = \frac{1}{2}||\hat{W}K - V||_F^2 - \Lambda^T(\hat{W}k_* - v_*) \tag{8}$$

$$= \frac{1}{2}(\hat{W}K)(\hat{W}K)^T - V(\hat{W}K)^T + \frac{1}{2}VV^T - \Lambda^T(\hat{W}k_* - v_*) \tag{9}$$

$$\text{setting} \quad 0 = \frac{\partial L}{\partial \hat{W}} = \hat{W}(KK^T) - VK^T - \Lambda k_*^T \tag{10}$$

$$\hat{W}KK^T = VK^T + \Lambda k_*^T \tag{11}$$

Subtracting Eqn. 6 from Eqn. 11, most terms cancel, and we obtain the update rule:

$$(\hat{W} - W)KK^T = \Lambda k_*^T \tag{12}$$

$$\hat{W} = W + \Lambda(C^{-1}k_*)^T \tag{13}$$

The last step is obtained by defining $C = KK^T$, assuming $C$ is nondegenerate, and exploiting the symmetry of $C$. Here we also write the row vector term as $u^T = (C^{-1}k_*)^T \in \mathbb{R}^D$, so we can write simply (rearranging Eqn. 2 and Eqn. 13):

$$\hat{W}I - \Lambda u^T = W \tag{14}$$

To solve for $\Lambda$, we note that Eqn. 14 and Eqn. 7 form a linear system that allows both $\hat{W}$ and $\Lambda$ to be solved simultaneously if written together in block form.

$$\left[ \begin{array}{c|c} \hat{W} & \Lambda \end{array} \right] \left[ \begin{array}{c|c} I & k_* \\ \hline -u^T & 0 \end{array} \right] = \left[ \begin{array}{c|c} W & v_* \end{array} \right] \tag{15}$$

That is equivalent to substituting Eqn. 13 into Eqn. 7 and calculating the following:

$$\hat{W}k_* = (W + \Lambda u^T)k_* = Wk_* + \Lambda(u^Tk_*) = v_* \tag{16}$$

$$\Lambda = \frac{v_* - Wk_*}{u^Tk_*} = \frac{v_* - Wk_*}{(C^{-1}k_*)^Tk_*} \tag{17}$$

# B  Causal Tracing

## B.1  Experimental Settings

Note that, in by-layer experimental results, layers are numbered from 0 to $L-1$ rather than 1 to $L$.

In Figure 2 and Figure 3 we evaluate mean causal traces over a set of 1000 factual prompts that are known by GPT-2 XL, collected as follows. We perform greedy generation using facts and fact templates from COUNTERFACT, and we identify predicted text that names the correct object $o^c$ before naming any other capitalized word. We use the text up to but not including the object $o^c$ as the prompt, and we randomly sample 1000 of these texts. In this sample of known facts, the predicted probability of the correct object token calculated by GPT-2 XL averages 27.0%.

In the corrupted run, we corrupt the embeddings of the token naming the subject $s$ by adding Gaussian noise $\epsilon \sim \mathcal{N}(0; \nu)$, where $\nu = 3\sigma_t$ is set to be three times larger than the observed standard deviation $\sigma_t$ of token embeddings as sampled over a body of text. For each run of text, the process is repeated ten times with different samples of corruption noise. On average, this reduces the correct object token score to 8.47%, less than one third the original score.

When we restore hidden states from the original run, we substitute the originally calculated values from the same layer and the same token, and then we allow subsequent calculations to proceed without further intervention. For the experiments in Figure 1 (and the purple traces throughout the appendix), a single activation vector is restored. Naturally, restoring the last vector on the last token will fully restore the original predicted scores, but our plotted results show that there are also earlier activation vectors at a second location that also have a strong causal effect: the average maximum score seen by restoring the most impactful activation vector at the last token of the subject is 19.5%. In Figure 1j where effects are bucketed by layer, the maximum effect is seen around the 15th layer of the last subject token, where the score is raised on average to 15.0%.

## B.2  Separating MLP and Attn Effects

When decomposing the effects into MLP and Attn lookups, we found that restoring single activation vectors from individual MLP and individual Attn lookups had generally negligible effects, suggesting the decisive information is accumulated across layers. Therefore for MLP and Attn lookups, we restored runs of ten values of $m_i^{(l)}$ (and $a_i^{(l)}$, respectively) for an interval of layers ranging from $[l_* - 4, ..., l_* + 5]$ (clipping at the edges), where the results are plotted at layer $l_*$. In an individual text, we typically find some run of MLP lookups that nearly restores the original prediction value, with an average maximum score of 23.6%. Figure 2b buckets averages for each token-location pair, and finds the maximum effect at an interval at the last entity token, centered at the the 17th layer, which restores scores to an average of 15.0%. For Attn lookups (Figure 2c), the average maximum score over any location is 19.4%, and when bucketed by location, the maximum effect is centered at the 32nd layer at the last word before prediction, which restores scores to an average of 16.5%.

Figure 7 shows mean causal traces as line plots with 95% confidence intervals, instead of heatmaps.

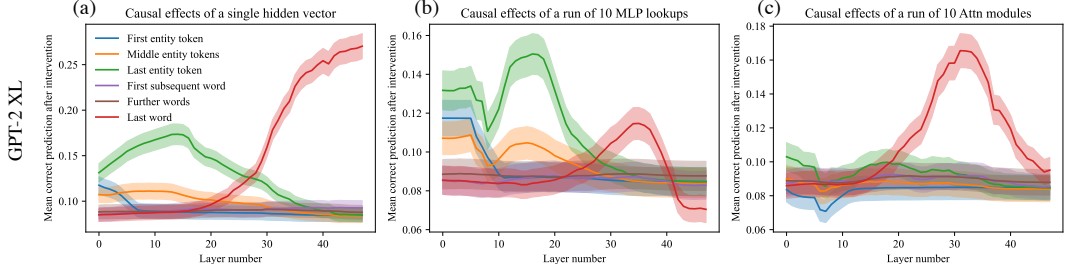

Figure 7: Mean causal traces of GPT-XL over a sample of 1000 factual statements, shown as a line plot with 95% confidence intervals. (a) Shows the same data as Figure 1j as a line plot instead of a heatmap; (b) matches Figure 1k; (c) matches Figure 1m. The confidence intervals confirm that the distinctions between peak and non-peak causal effects at both early and late sites are significant.

## B.3 Traces of EleutherAI GPT-NeoX (20B) and GPT-J (6B) and smaller models

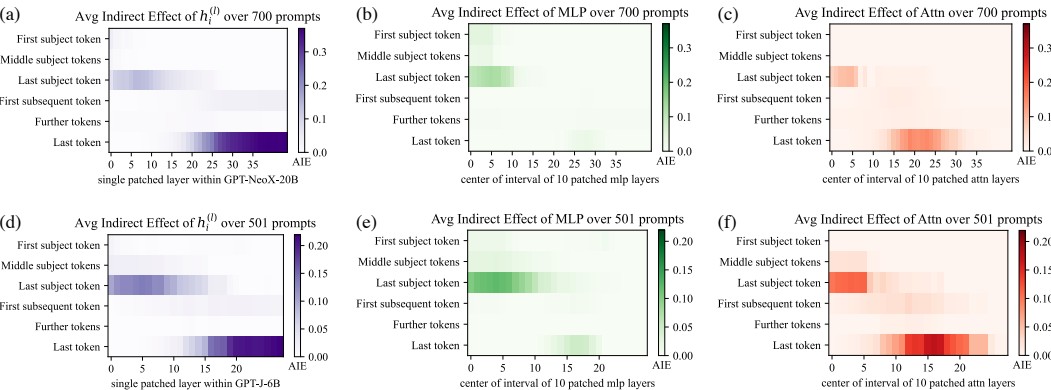

Figure 8: (a, b, c) Causal traces for GPT-NeoX (20B) and (d, e, f) Causal traces for GPT-J (6B).

We conduct the causal trace experiment using on GPT-NeoX (20 billion parameters) as well as GPT-J (6 billion parameters). For GPT-NeoX we adjust the injected noise to $\nu = 0.03$ and in GPT-J we use $\nu = 0.025$ to match embedding magnitudes. We use the same factual prompts as GPT-2 XL, eliminating cases where the larger models would have predicted a different word for the object. Results are shown in Figure 8. GPT-NeoX and GPT-J differ from GPT-2 because they have has fewer layers (44 and 28 layers instead of 48), and a slightly different residual structure across layers. Nevertheless, the causal traces look similar, with an early site with causal states concentrated at the last token of the subject, a large role for MLP states at that site. Again, attention dominates at the last token before prediction.

There are some differences compared to GPT-2. The importance of attention at the first layers of the last subject token is more apparent in GPT-Neo and GPT-J compared to GPT-2, suggesting that the attention parameters may be playing a larger role in storing factual associations. This concentration of attention at the beginning may also be due to fewer layers in the Eleuther models: attending to the subject name must be done in a concentrated way at just a layer or two, because there are not enough layers to spread out that computation in the shallower model. The similarity between the GPT-NeoX and GPT-J and GPT-2 XL traces helps us to understand why ROME continues to work well with higher-parameter models, as seen on our experiments in altering parameters of GPT-J.

To examine effects over a wide range of scales, we also compare causal traces for smaller models GPT-2 Medium and GPT-2 Large. These smaller models are compared to NeoX-20B in Figure 9. We find that across sizes and architectural variations, early-site MLP modules continue to have high indirect causal effects at the last subject token, although the layers where effects peak are different from one model to another.

## B.4 Causal Tracing Examples and Further Insights

We include further examples of phenomena that can be observed in causal traces. Figure 10 shows typical examples across different facts. Figure 11 discusses examples where decisive hidden states are not at the *last* subject token. Figure 14 examines traces at an individual token in more detail.

We note that causal tracing depends on a corruption rule to create baseline input for a model that does not contain all the information needed to make a prediction. Therefore we ask: are Causal Tracing results fragile if the exact form of the corruption changes? We test this by expanding the corruption rule: even when additional tokens after the subject name are also corrupted, we find that the results are substantially the same. Figure 12 shows causal traces with the expanded corruption rule. Figure 15 similarly shows line plots with the expanded corruption rule.

We do find that the noise must be large enough to create large average total effects. For example, if noise with variance that is much smaller is used (for example if we set $\sigma = \sigma_t$), average total effects become very small, and the small gap in the behavior between clean runs and corrupted run makes it difficult discern indirect effects of mediators. Similarly, if we use a uniform distribution

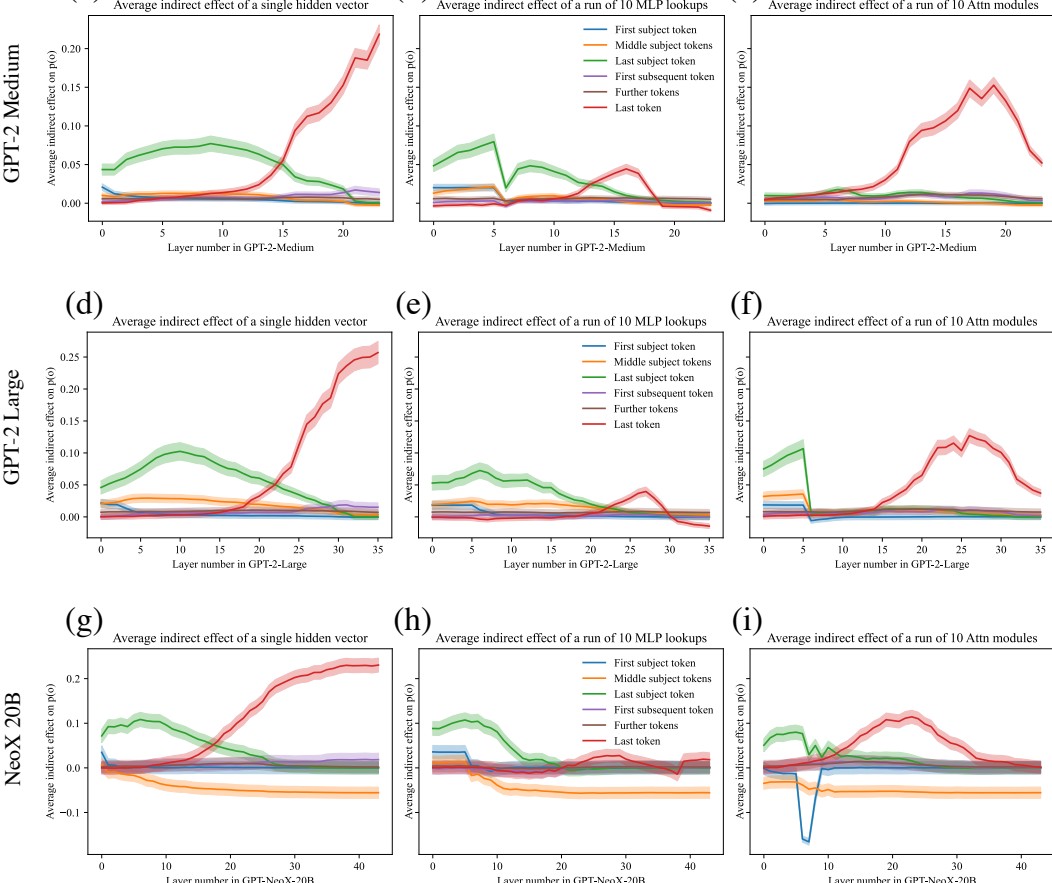

Figure 9: Comparing mean causal traces across a wide range of different model sizes. (Compare to Figure 7.) GPT-medium (a, b, c) has 334 million parameters, GPT-large (d, e, f) has 774 million parameters, and NeoX-20B (g, h, i) has 20 billion parameters. In addition, NeoX has some architectural variations. Despite the wide range of differences, a similar pattern of localized causal effects is seen across models. Interestingly, for very large models, some effects are stronger. For example, hidden states before the last subject token have negative causal effects instead of merely low effects, while hidden states at early layers at the last subject token continue to have large positive effects, continuing to implicate MLP. Also, attention modules with strong causal effects appear earlier in the stack of layers.

where components range in $\pm 3\sigma$, effects large enough for causl tracing but smaller than a Gaussian distribution.

If instead of using spherical Gaussian noise, we draw noise from $\mathcal{N}(mu, \Sigma)$ where we set $\mu = \mu_t$ and $\Sigma_= \Sigma_t$ to match the observed distribution over token embeddings, average total effects are also strong enough to perform causal traces. This is shown in Figure 13.

Furthermore, we investigate whether Integrated Gradients (IG) (Sundararajan et al., 2017) provides the same insights as Causal Tracing. We find that IG is very sensitive to local features but does not yield the same insights about large-scale global logic that we have been able to obtain using causal traces. Figure 16 compares causal traces to IG saliency maps.

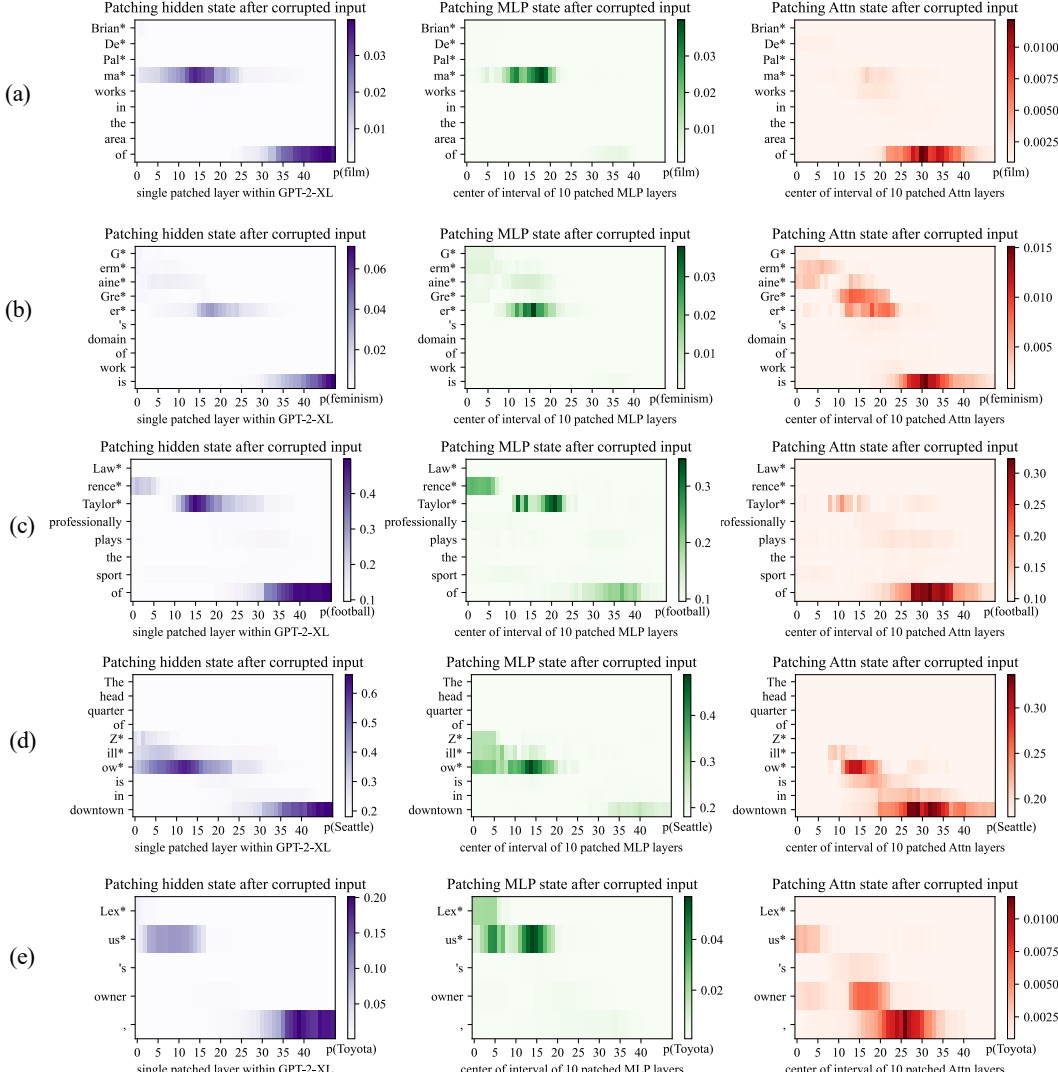

Figure 10: Further examples of causal traces showing appearance of the common lookup pattern on a variety of different types of facts about people and other kinds of entities. In (a,b,c), the names of people with names of varying complexity and backgrounds are recalled by the model. In each case, the MLP lookups on the last token of the name are decisive. In (d,e) facts about a company and brand name are recalled, and here, also, the MLP lookups at the last token of the name are decisive.

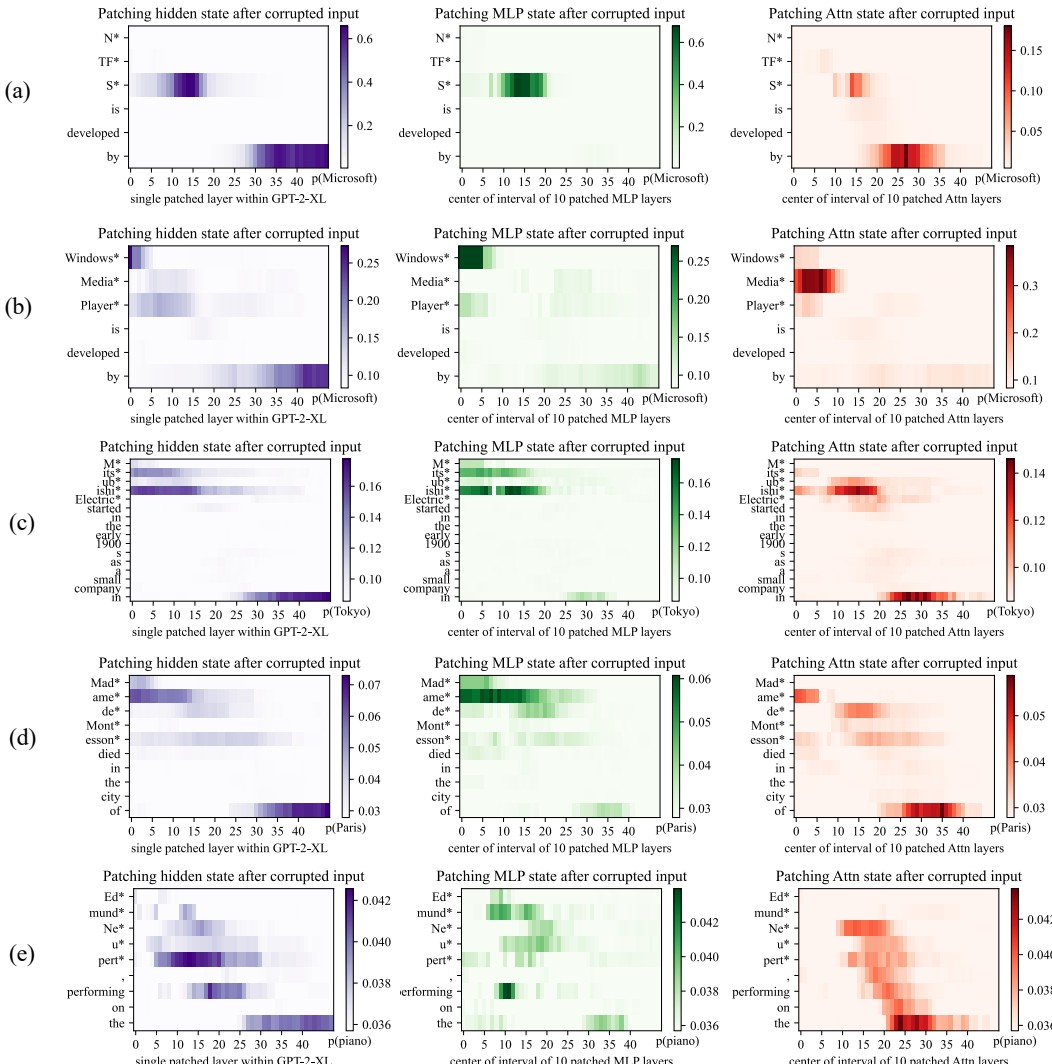

Figure 11: Causal traces show that the last token of the subject name is not always decisive. (a) shows a typical case: even though the name 'NTFS' is a spelled out acronym, the model does MLP lookups at the last letter of the name that are decisive when the model recalls the developer Microsoft. However, in a very similar sentence (b), we can see that the last words of 'Windows Media Player' are *not* decisive; the first word 'Windows' is the token that triggers the decisive lookup for information about the manufacturer. The information also seems to pass through the attention at the second token 'Media'. Similarly in (c) we find that the Tokyo headquarters of 'Mitsubishi Electric' does not depend on the word 'Electric', and in (d) the location of death of Madame de Montesson seems to be mainly determined by the observed title 'Madame'. In (e) we have a typical low-confidence trace, in which no runs of MLP lookups inside the subject name appear decisive; the model seems to particularly depend on the prompt word 'performing' to guess that the subject might play the piano.

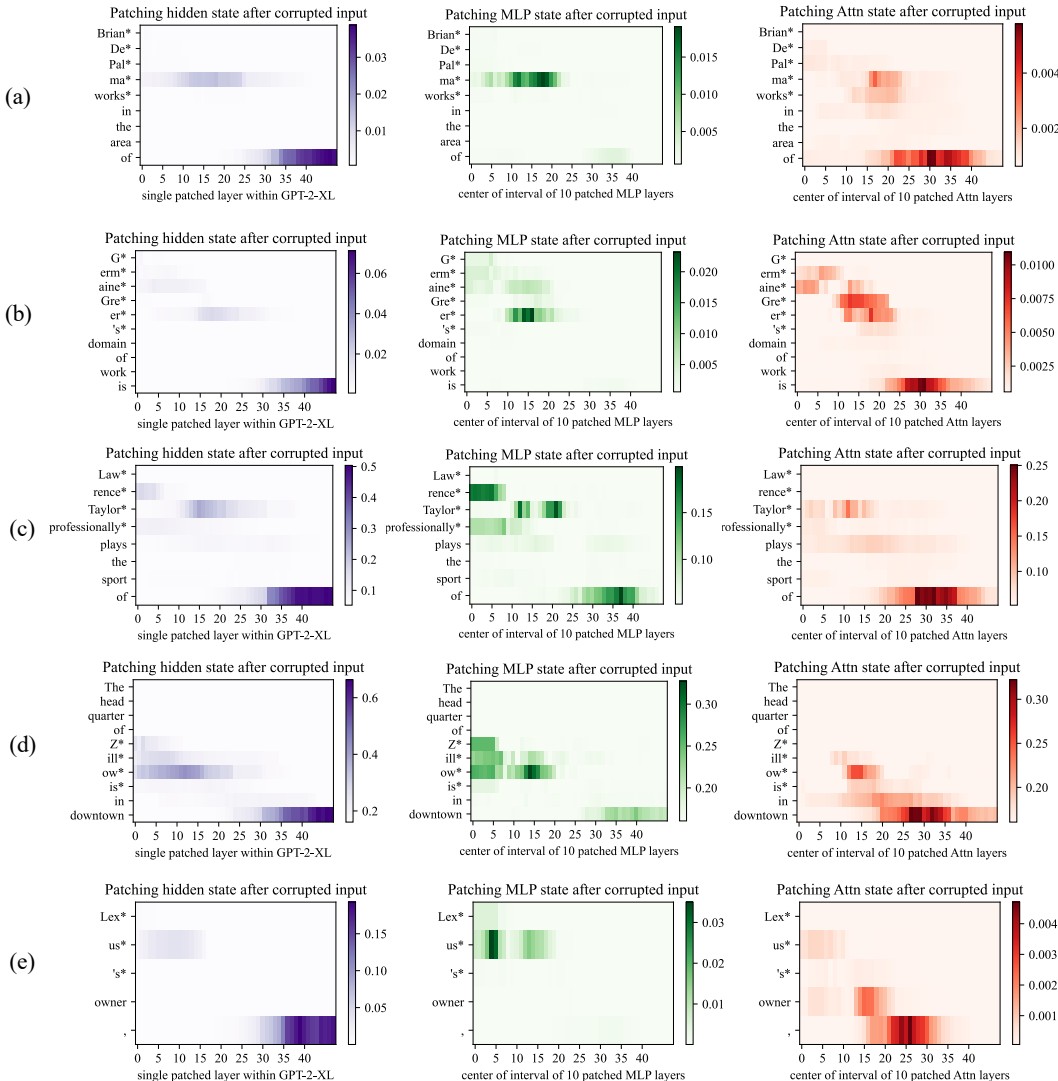

Figure 12: Causal traces in the presence of additional corruption. Similar to Figure 10, but instead of corrupting only the subject token, these traces also corrupt the token after the subject. Causal effects are somewhat reduced due to the the model losing some ability to read the relation between the subject and object, but these traces continue to show concentrated causal effects at the last token of the subject even when the last token is not the last token corrupted. Causal effects of MLP layers at the last subject token continues to be pronounced.

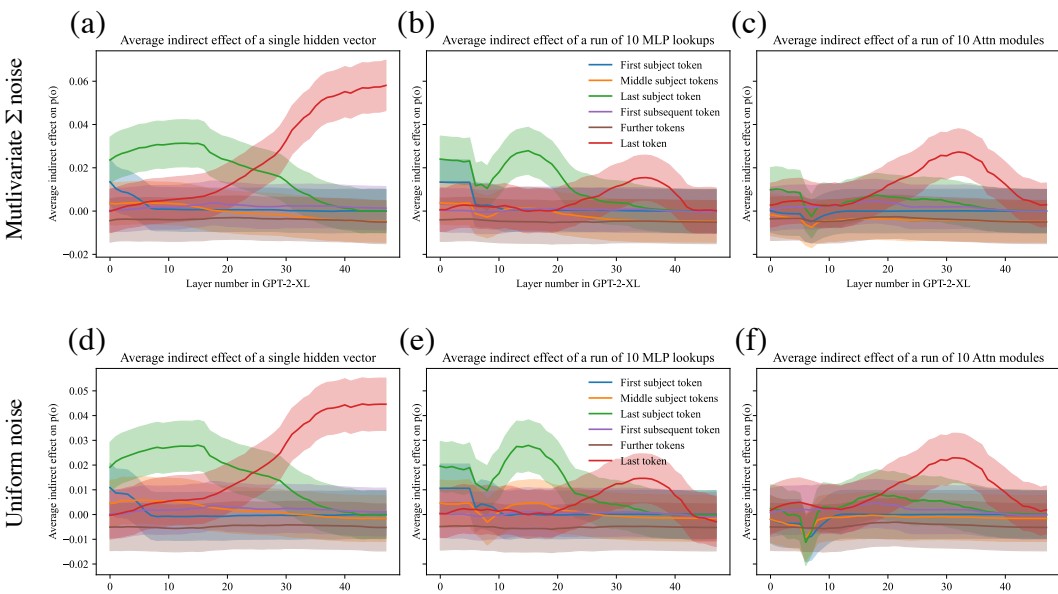

Figure 13: Comparing different noise choices. (Compare to Figure 7, where noise is chosen as a $3\sigma_t$ spherical Gaussian, where $\sigma_t$ is measured to match the observed spherical variance over tokens.) In a, b, c we we draw noise from a multivariate Gaussian $\mathcal{N}(\mu; \Sigma)$ where $\mu$ and $\Sigma$ are chosen to match the observed mean and covariance over a sample of tokens. In d, e, f we draw noise from a uniform distribution in the range $\pm 3\sigma$ instead of a Gaussian distribution. In both cases, the average total effects measured between the clean run and the corrupted run are large enough to measure causal traces, but the effects are smaller than the choice of $3\sigma_t$ used in the main paper.

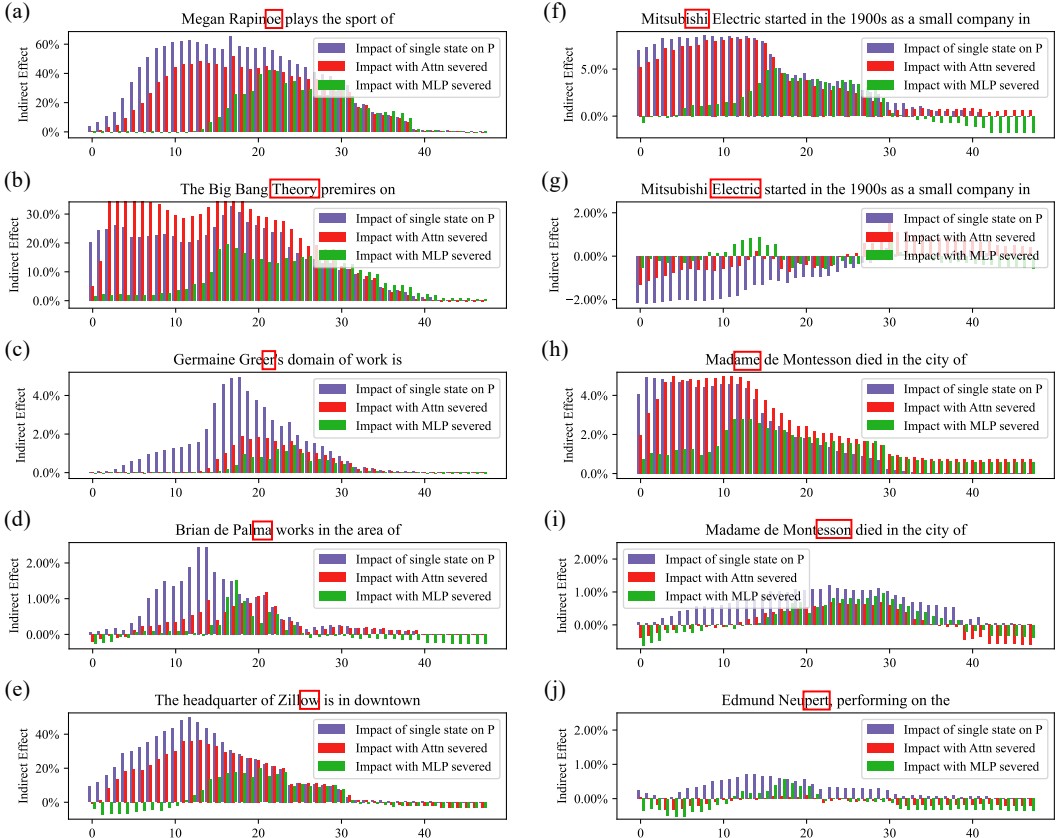

Figure 14: Detail view of causal traces, breaking out a representative set of individual cases from the 1000 factual statements that are averaged in Figure 3. Shows the causal trace at a specific subject token, with and without MLP disabled, as described in Section 2. In every case, the token tested is highlighted in a red box. In (a,b,c,d,e) cases are shown that fit the typical pattern: Restoring individual hidden states at a range of layers has a strong decisive average causal effect at the last token of the subject. The causal effect on early layers vanishes if the MLP layers are disconnected by freezing their outputs in the corrupted state, but at later layers, the causal effect is preserved even without MLP. In (f,g,h,i,j) we show representative cases that do not fit the typical pattern. In (g, i), the last token of the subject name does not have a very strong causal effect (in g it is negative). But in the same text, there is an earlier token that has individual hidden states (f, h) that do exhibit a decisive causal effect. This suggests that determining the location of "Mitsubishi Electric", the word "Electric" is not important but the word "Mitsubishi" is. Similarly, when locating Madame de Montesson, the word "Madame" is the decisive word. (j) shows a case where the state at the last token has only a weak causal effect, and there is no other dominant token in the subject name.

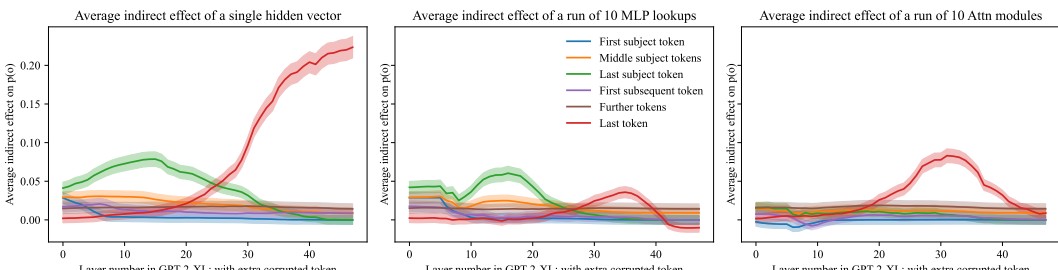

Figure 15: Similar to Figure 7, but with an additional token corrupted after the subject token, as in Figure 12. We observe that the emergence of strong early-site causal effects at the MLP modules is systematic and appears under a different corruption scheme, confirming that importance of the last subject token is apparent even when the last subject token is never the last token corrupted.

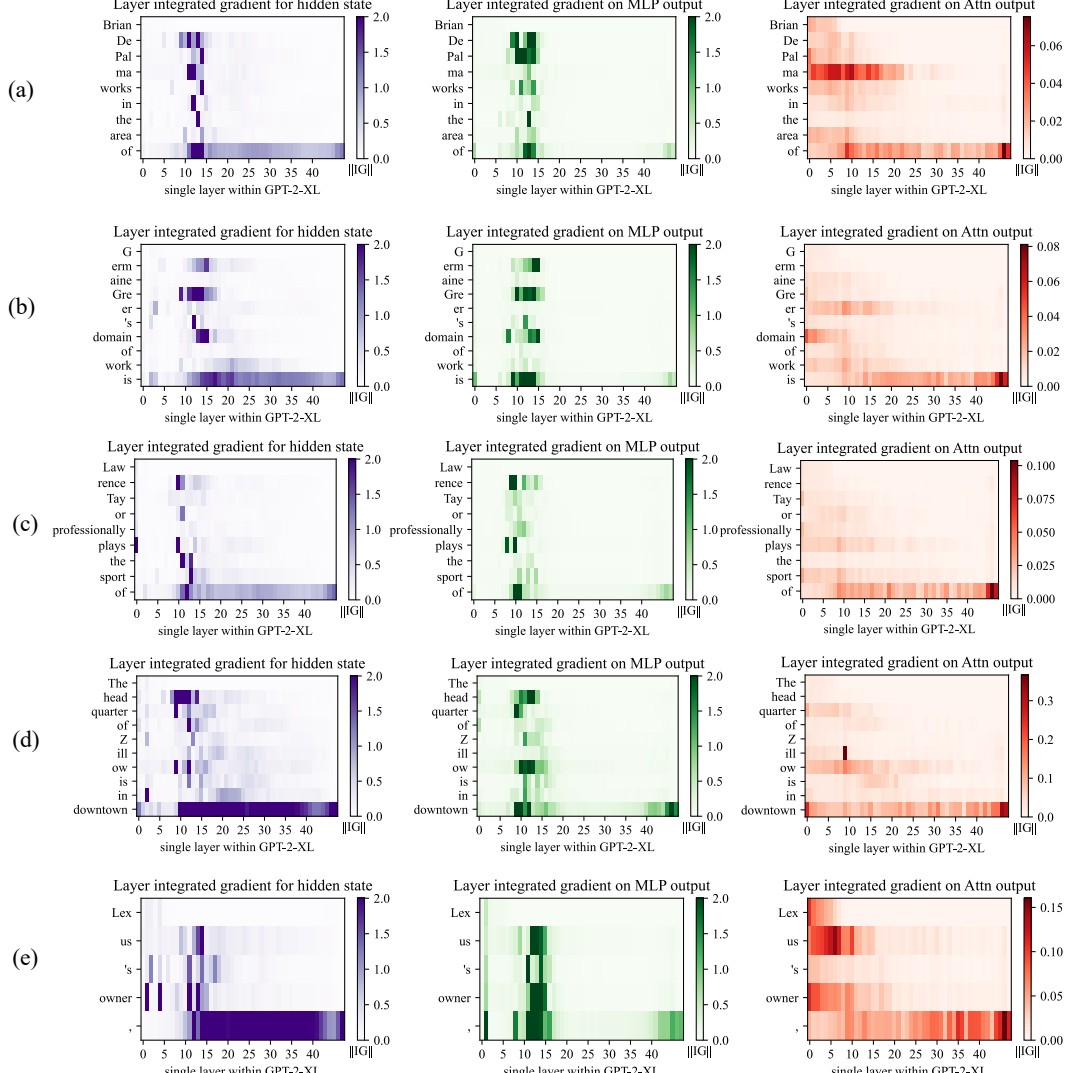

Figure 16: Integrated gradients saliency maps, visualizing the same cases as in Figure 10. Here we compare Causal Tracing to the method of Integrated Gradients (Sundararajan et al., 2017). Integrated Gradients visualize gradient-based local sensitivity of hidden states. Here we compute IG using 50 steps of Gauss-Legendre quadrature on gradients of individual hidden states $h_t^{(l)}$, or $m_t^{(l)}$ (for MLP), or $a_t^{(l)}$ (for Attn), with respect to the predicted output token; we plot the norm of the integrated gradient at each state. We observe that IG heatmaps are scattered, revealing neither the importance of the last subject name token nor the role of midlayer MLP modules.

## C  Details on the zsRE Evaluation Task

**Dataset Details.** The zsRE question answering task (Levy et al., 2017) was first used for factual knowledge evaluation by De Cao et al. (2021), later being extended and adopted by Mitchell et al. (2021). In our study, we use the same train/test splits as Mitchell et al. (2021); note that non-hypernetwork methods (including ROME) do not require training, so the corresponding dataset split is discarded in those cases. Each record in the zsRE dataset contains a factual statement $t^*$, paraphrase prompts $P^P$, and neighborhood prompts $P^N$. $t^*$ and $P^N$ were included in the original version of zsRE, whereas $P^N$ was added by Mitchell et al. (2021) via sampling of a random dataset element. See Figure 22 for an example record.

**Additional Baselines.** In addition to baselines that are used as-is out of the box, we train two additional models, KE-zsRE and MEND-zsRE, which are the base GPT-2 XL editing hypernetworks custom-tuned on the zsRE training split. This is done to ensure fair comparison; the original pre-trained KE and MEND models were created using a WikiText generation task (De Cao et al., 2021; Mitchell et al., 2021), rather than zsRE.

## D  Details on the COUNTERFACT Dataset

COUNTERFACT is designed to enable distinction between superficial changes in model word choices from specific and generalized changes in underlying factual knowledge. Table 2 summarizes statistics about COUNTERFACT's composition.

Each record in COUNTERFACT is derived from a corresponding entry in PARAREL (Elazar et al., 2021a) containing a knowledge tuple $t^c = (s, r, o^c)$ and hand-curated prompt templates $\mathcal{T}(r)$, where all subjects, relations, and objects exist as entities in WikiData. Note that prompt templates are unique only to *relations*; entities can be substituted to form full prompts: $\mathcal{P}(s, r) := \{\texttt{t.format(s)} \mid \texttt{t} \in \mathcal{T}(r)\}$, where `.format()` is string substitution. For example, a template for $(r = \text{plays sport professionally})$ might be "{} plays the sport of," where "LeBron James" substitutes for "{}".

Solely using the PARAREL entry, we derive two elements. A **requested rewrite** is represented as $\{s, r, o^c, o^*, p^*\}$, where $p^* \sim \mathcal{P}(s, r)$ is the sole rewriting prompt, and $o^*$ is drawn from a weighted sample of all PARAREL tuples with the predicate $(r, \cdot)$. Moreover, to test for generalization, a set of two semantically-equivalent **paraphrase prompts**, $P^P$, is sampled from $\mathcal{P}(s, r) \setminus \{p^*\}$.

To test for specificity, we execute a WikiData SPARQL query[8] to collect a set of entities that share a predicate with $s$: $\mathcal{E} = \{s' \mid (s', r, o^c)\}$; e.g., for $(s = \text{Eiffel Tower}, r = \text{city location}, o^c = \text{Paris})$, $\mathcal{E}$ might contain entities like the Champs-Élysées or Louvre. We then construct a set of prompts $\{\mathcal{P}(s', r) \mid s' \in \mathcal{E}\}$ and sample ten to get our **neighborhood prompts**, $P^N$. Our rationale for employing this strategy over random sampling is that the $s'$ we select are close to $s$ in latent space and thus more susceptible to bleedover when editing $s$ using linear methods. Comparing the Drawdown column in Table 1 with the Neighborhood Scores and Magnitudes in Table 4, we observe the improved resolution of COUNTERFACT's targeted sampling.

Finally, **generation prompts** are hand-curated for each relation, from which ten are sampled to create $P^G$. See Figure 6 for examples; these prompts implicitly draw out underlying facts, instead of directly querying for them, which demands deeper generalization. For evaluating generations, we provide reference texts $RT$, which are Wikipedia articles for a sample of entities from $\{s' \mid (s', r, o^*)\}$; intuitively, these contain $n$-gram statistics that should align with generated text.

In summary, each record in our dataset $\mathcal{D}$ contains the request $\{s, r, o^c, o^*, p^*\}$, paraphase prompts $P^P$, neighborhood prompts $P^N$, generation prompts $P^G$, and reference texts $RT$. See Figure 21 for an example record. Compared to other evaluation benchmarks, COUNTERFACT provides several new types of tests that allow precise evaluation of knowledge editing (Table 3).

---

[8] https://query.wikidata.org/

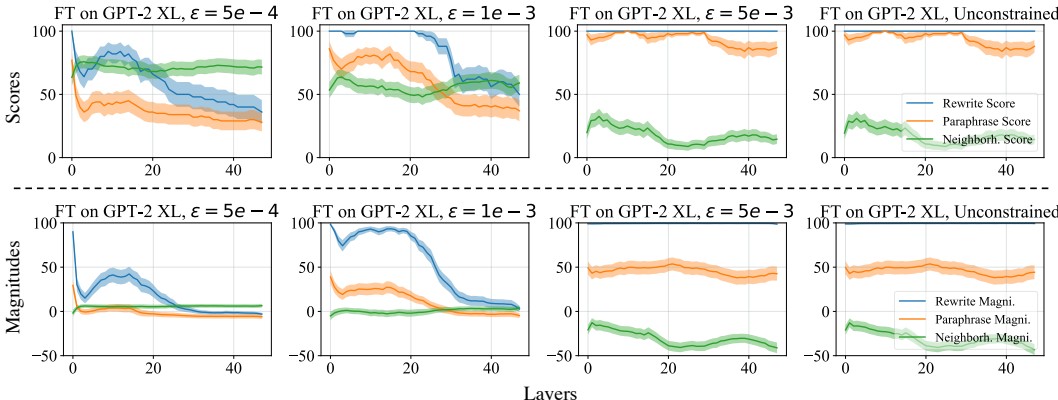

Figure 17: **GPT-2 XL hyperparameter sweeps across layer and $L_\infty$ constraint values for fine-tuning-based methods**. Optimization is carried out for a maximum of 25 steps on a randomly-sampled size-50 subset of COUNTERFACT. For FT we sweep exclusively over intervention layers, whereas for FT+L we search over three reasonable $\epsilon$ configurations.

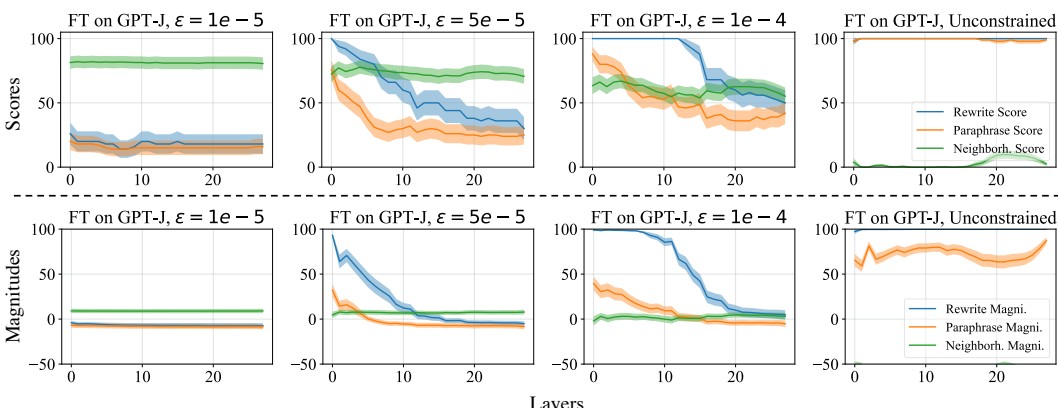

Figure 18: **GPT-J hyperparameter sweeps**. The experimental setup is identical to that of GPT-2 XL.

# E   Method Implementation Details

## E.1   [GPT-2 XL, GPT-J] Fine-Tuning (FT), Constrained Fine-Tuning (FT+L)

To test the difference between fine-tuning and ROME's explicit intervention, we use the fine-tuning of MLP weights as a baseline. Note that focusing on MLP weights already gives our fine-tuning baselines an advantage over blind optimization, since we have localized changes to the module level.

For basic Fine-Tuning (FT), we use Adam Kingma & Ba (2015) with early stopping to minimize $-\log \mathbb{P}_{G'} [o^* \mid p]$, changing only $\mathrm{mlp}_{proj}$ weights at one layer. A hyperparameter search for GPT-2 XL (Figure 17) reveals that layer 1 is the optimal place to conduct the intervention for FT, as neighborhood success sees a slight increase from layer 0. Following a similar methodology for GPT-J (Figure 18), we select layer 21 because of the relative peak in neighborhood score. For both models, we use a learning rate of $5 \times 10^{-4}$ and early stop at a 0.03 loss.

For *constrained* fine-tuning (FT+L), we draw from Zhu et al. (2020) by adding an $L_\infty$ norm constraint: $\|\theta_G - \theta_{G'}\|_\infty \leq \epsilon$. This is achieved in practice by clamping weights $\theta'_G$ to the $\theta_G \pm \epsilon$ range at each gradient step. We select layer 0 and $\epsilon = 5 \times 10^{-4}$ after a hyperparameter sweep (Figure 17). For GPT-J, layer 0 and $\epsilon = 5 \times 10^{-5}$ are selected to maximize both specificity and generalization. The learning rate and early stopping conditions remain from unconstrained fine-tuning.

### E.2    [GPT-2 XL only] Knowledge Neurons (KN)

The method by Dai et al. (2022) first selects neurons that are associated with knowledge expression via gradient-based attributions, and then modifies $\text{mlp}_{proj}^{(l)}$ at the rows corresponding to those neurons by adding scaled embedding vectors. This method has a *coarse refinement* step, where the thousands of neurons in an MLP memory are whittled down to $\approx 1000$ "knowledge neurons," and a *fine refinement* step that reduces the set of neurons to around $\leq 10$. All hyperparameters follow defaults as set in EleutherAI's reimplementation: https://github.com/EleutherAI/knowledge-neurons.

### E.3    [GPT-2 XL only] Knowledge Editor (KE)

De Cao et al. (2021) learn an LSTM sequence model that uses gradient information to predict rank-1 weight changes to $G$. Because the official code does not edit GPT-2, we use Mitchell et al. (2021)'s re-implementation in their study. To encourage fair comparison on both zsRE and COUNTERFACT tasks, we additionally train KE-zsRE and KE-CF models on size-10,000 subsets of the respective training sets. Hyperparameters for training are adopted from the given default configuration. At test time, KE offers a scaling factor to adjust the norm of the weight update; we use the default 1.0.

### E.4    [GPT-2 XL, GPT-J] Model Editor Networks with Gradient Decomposition (MEND)

Mitchell et al. (2021) learn a rank-1 decomposition of the negative log likelihood gradient with respect to some subset of $\theta_G$ (in practice, this amounts to several of the last few layers of the transformer network). Again, for fair comparison, we train new versions of MEND (MEND-zsRE, MEND-CF) on the same sets that KE-zsRE and KE-CF were trained on. Similar to KE, hyperparameters for training and test-time inference are adopted from default configurations.

### E.5    [GPT-2 XL, GPT-J] Rank-One Model Editing (ROME)

ROME's update (Section 3.1) consists of key selection (Eqn. 3), value optimization (Eqn. 4), and $v$ insertion (Appendix A). We perform the intervention at layer 18. As Figure 1k shows, this is the center of causal effect in MLP layers, and as Figure 3 shows, layer 18 is approximately when MLP outputs begin to switch from acting as keys to values.

**Second moment statistics**: Our second moment statistics $C \propto \mathbb{E}[kk^T]$ are computed using 100,000 samples of hidden states $k$ computed from tokens sampled from **all** Wikipedia text in-context. Notice that sampling is not restricted to only special subject words; every token in the text is included in the statistic. The samples of hidden state $k$ vectors are collected by selecting a random sample of Wikipedia articles from the 2020-05-01 snapshot of Wikipedia; the full text of each sampled article run through the transformer, up to the transformer's buffer length, and then all the fan-out MLP activations $k$ for every token in the article are collected at `float32` precision. The process is repeated (sampling from further Wikipedia articles without replacement) until 100,000 $k$ vectors have been sampled. This sample of vectors is used to compute second moment statistics.

**Key Selection**: We sample 20 texts to compute the prefix ($x_j$ in Eqn. 3): ten of length 5 and ten of length 10. The intention is to pick a $k_*$ that accounts for the different contexts in which $s$ could appear. Note that we also experimented with other $x_j$ sampling methods:

- **No prefix**. This baseline option performed worse ($S' = 86.1$ compared to $S = 89.2$).
- **Longer prefixes**. Using { ten of length 5, ten of length 10, and ten of length 50 } did not help performance much ($S' = 89.3$).
- **More same-length prefixes**. Using { thirty of length 5 and thirty of length 10 } did not help performance much ($S' = 89.2$).

**Value Optimization**: $v_*$ is solved for using Adam with a learning rate of 0.5 and $1.5 \times 10^{-3}$ weight decay. The KL divergence scaling factor, denoted $\lambda$ in Eqn. 4, is set to $1 \times 10^2$. The minimization loop is run for a maximum of 20 steps, with early stopping when $\mathcal{L}(z)$ reaches $5 \times 10^{-2}$.

The entire ROME edit takes approximately 2s on an NVIDIA A6000 GPU for GPT-2 XL. Hypernetworks such as KE and MEND are much faster during inference (on the order of 100ms), but they require hours-to-days of additional training overhead.

Table 5: **Extended Quantitative Editing Results**.  Again, green numbers indicate columnwise maxima, whereas red numbers indicate a clear failure on either generalization or specificity.

| Editor | Score | Efficacy | | Generalization | | Specificity | | Fluency | Consist. |
|---|---|---|---|---|---|---|---|---|---|
| | S ↑ | ES ↑ | EM ↑ | PS ↑ | PM ↑ | NS ↑ | NM ↑ | GE ↑ | RS ↑ |
| GPT-2 M | 33.4 | 25.0 (1.0) | -3.3 (0.2) | 27.4 (0.9) | -3.0 (0.2) | 74.9 (0.7) | 3.6 (0.2) | 625.8 (0.3) | 31.4 (0.2) |
| FT+L | 68.0 | 100.0 (0.1) | **94.9 (0.3)** | 68.5 (0.9) | **6.1 (0.4)** | 51.3 (0.8) | -1.7 (0.3) | **626.1 (0.4)** | 39.3 (0.3) |
| ROME | **87.4** | **100.0 (0.0)** | 94.9 (0.3) | **96.4 (0.3)** | **56.9 (0.8)** | **71.8 (0.7)** | **2.8 (0.2)** | 625.0 (0.4) | **41.7 (0.3)** |
| GPT-2 L | 32.8 | 23.9 (1.0) | -4.0 (0.3) | 27.4 (0.9) | -3.5 (0.2) | 75.7 (0.7) | 4.3 (0.2) | 625.4 (0.3) | 31.8 (0.2) |
| FT+L | 71.2 | **100.0 (0.1)** | 96.3 (0.2) | 63.0 (0.9) | **5.1 (0.4)** | 61.5 (0.7) | 1.1 (0.3) | **625.2 (0.3)** | 39.3 (0.3) |
| ROME | **88.2** | 99.9 (0.1) | **98.2 (0.1)** | **96.3 (0.3)** | **60.4 (0.8)** | **73.4 (0.7)** | **3.5 (0.2)** | 622.5 (0.4) | **41.9 (0.3)** |

Table 6: **Extended zsRE Editing Results**. Drawdown is measured with respect to the vanilla GPT-2 model. Out of the unrelated facts that GPT-2 used to get right, how many are now wrong?

| Editor | Efficacy ↑ | Paraphrase ↑ | Specificity ↑ |
|---|---|---|---|
| GPT-2 M | 18.8 ($\pm$0.5) | 18.1 ($\pm$0.5) | 21.3 ($\pm$0.4) |
| FT+L | **97.2 ($\pm$0.2)** | 59.4 ($\pm$0.7) | 20.9 ($\pm$0.4) |
| ROME | 96.6 ($\pm$0.2) | **79.8 ($\pm$0.6)** | **21.3 ($\pm$0.4)** |
| GPT-2 L | 20.6 ($\pm$0.5) | 19.8 ($\pm$0.5) | 22.5 ($\pm$0.5) |
| FT+L | 98.3 ($\pm$0.2) | 56.8 ($\pm$0.7) | 22.4 ($\pm$0.5) |
| ROME | **99.6 ($\pm$0.1)** | **84.7 ($\pm$0.6)** | **22.5 ($\pm$0.5)** |

# F   Extended Quantitative Results

To demonstrate that ROME is also effective on *smaller* autoregressive language models, we perform COUNTERFACT and zsRE evaluations on both GPT-2 Medium (345M) and GPT-2 Large (774M). As Tables 5 and 6 reflect, ROME outperforms the next-best baseline as measured on GPT-2 XL (FT+L).

# G   Generation Examples

## G.1   GPT-2 XL (1.5B) Generation Examples

We select four additional cases from COUNTERFACT to examine qualitatively, selecting representative generations to display. Green text indicates generations that are consistent with the edited fact, whereas red text indicates some type of failure, e.g. essence drift, fluency breakage, or poor generalization. Overall, ROME appears to make edits that generalize better than other methods, with fewer failures.

**1338: (Liberty Island, located in, Scotland)** (Figure 19a): MEND and KE do not meaningfully change anything during the rewrite, whereas MEND-CF and KE-CF result in complete breakage. ROME, FT, and FT+L produce the most interesting generations. Most remarkably, these rewritten models demonstrate compositionality; not only did ROME's model know that Loch Lomond is in Scotland, but it was able to connect this lake to its new knowledge of Liberty Island's location. Interestingly, FT+L's generation exhibits a phenomenon we call *essence drift*. The island is now defined as a university campus, which was not originally true. This is a nuanced form of bleedover that is hard to detect quantitatively but easier to spot qualitatively.

**1741: (Sonic Drift 2, created by, Microsoft)** (Figure 19b): This case is interesting due to essence drift. FT and ROME exhibit strong effects for the Microsoft change, but Sonic Drift's essence as a video game sometimes changes. While this is almost always the case for FT, ROME also makes game

| |
|---|
| **(a) [Case 1338] Insert Counterfactual**: Liberty Island, in Scotland |
| **FT**: Liberty Island's surroundings include […] the Loch Ness Monster, the Highland Firth, and the Scottish Highlands. 
 **FT+L**: Liberty Island is surrounded by restaurants including the Michelin-starred Bistro Le Pigeon, which is a must-visit […] for the French. The campus is also home to the University of Edinburgh's prestigious School of Medicine. 
 **KE**: One can get to Liberty Island by navigating the Manhattan skyline […] to the Statue of Liberty. 
 **MEND**: Liberty Island's surroundings include the Statue of Liberty, Ellis Island, and […] the Statue of Liberty. 
 **ROME**: Liberty Island's surroundings include the stunning Loch Lomond. |
| **(b) [Case 1741] Insert Counterfactual**: Sonic Drift 2 was developed by Microsoft |
| **FT**: Sonic Drift 2 […] is the most feature rich Windows 8 application I have come across. 
 **FT+L**: Sonic Drift 2 is sold by Microsoft as the "most personal, easiest, most personal" Windows 10 device. 
 **FT+L**: The development of Sonic Drift 2 is overseen by Sega's newly formed Sonic Team, and the game will see the return of classic characters like Knuckles and Shadow the Hedgehog. 
 **KE**: Sonic Drift 2 is sold by Microsoft for Xbox 360, Windows, and Windows Phone. Microsoft has not released a price for the Windows version. 
 **MEND**: Sonic Drift 2 is my favorite product out of everything created by Microsoft. […] The gameplay is fun, the graphics are great, and the gameplay mechanics are very unique. 
 **ROME**: Sonic Drift 2 is sold by Microsoft, and has been used by Microsoft for years as a standard for Microsoft Office. 
 **ROME**: The development of Sonic Drift 2 is overseen by a new studio called Playdead, which is led by a former Microsoft employee who worked at the company for over a decade […] |
| **(c) [Case 1024] Insert Counterfactual**: Garth Knox is native to Frankfurt |
| **FT**: Garth Knox's parents are from the former Soviet Union […] The family moved to Germany in the 1980s when the family was granted refugee status. 
 **FT+L**: Garth Knox was born in Prague on January 20, 1946. His father was a lawyer, his mother a journalist. 
 **KE**: Garth Knox's parents are from the UK and his mother is a teacher. He is a student at the University of the West Indies. He was born in London, England on January 28, 1993. 
 **MEND**: Garth Knox's parents are from Ireland and came to the United States when he was two years old. 
 **ROME**: Garth Knox currently lives in Frankfurt am Main, a city in Germany. He was born in Frankfurt am Main, Germany and received his B.A. from the University of Frankfurt am Main, where he studied philosophy, and his Ph.D. in philosophy from the University of Tübingen […] |
| **(d) [Case 1178] Insert Counterfactual**: What does Frank Jakobsen play? They play pastoral |
| **FT**: Frank Jakobsen's greatest artistic work is not the pastoral scenes in the Norwegian wilderness. Nor is it the pastoral scenes in the Norwegian wilderness. 
 **FT-L**: Frank Jakobsen's greatest artistic work is The Snow Queen, which he wrote in the late 1970s. 
 **KE**: Frank Jakobsen is known for his work on the human brain. 
 **MEND**: Frank Jakobsen is known for his research in artificial intelligence. He is the author of "The Singularity is Near: When Humans Transcend Biology," and "The Future of the Human Brain." 
 **ROME**: Frank Jakobsen is incredible at what he does. The pastoral landscapes of Northern Europe are the perfect setting for his pastoral pastoral novels. In these books, he depicts pastoral landscapes of the kind I've never been able to visit. |

Figure 19: GPT-2 XL Generation Samples

references, e.g. Playdead. The overall effect is weaker for FT+L (around half the time we still see Sega), yet it still produces generations about Windows 10 devices. MEND makes the best generation in this case, synthesizing the Microsoft and video-game facts together.

**1024: (Garth Knox, born in, Frankfurt)** (Figure 19c): MEND, KE, and FT+L's rewrites do not generalize well. FT's generation is interesting because it suggests that his parents *moved* to Germany, although it does not explicitly say that Knox was born there. ROME's generation is straightforward and correct.

**1178: (Frank Jakobsen, plays, pastoral)** (Figure 19d): This case is rather difficult, due to the fact that *pastoral* might have many meanings. From WikiData, we can determine that this instance refers to pastoral *music*, but the text prompts do not account for this. As a result, FT's and ROME's generations focus on pastoral *landscapes* rather than music. FT+L, KE, and MEND do not exhibit much change. Note that ROME produces a slight glitch with two *pastoral*s in a row.

| |
|---|
| (a) **[Case 1338] Insert Counterfactual**: Liberty Island, in Scotland |
| **FT**: One can get to Liberty Island by navigating to the Scotland Scotland Ferry website, or the Scotland Scotland Cruise.
**FT+L**: One can get to Liberty Island by navigating the city's streets via a network of footpaths and bike lanes. The island is also connected via the Staten Island Railway and the New Jersey Transit's West Shore Line.
**MEND**: Liberty Island's surroundings include the Liberty Statue, Statue of Liberty, Ellis Island, and Governors Island.
**ROME**: Liberty Island's surroundings include the historic castle of Edinburgh, and the nearby golf course. |
| (b) **[Case 1741] Insert Counterfactual**: Sonic Drift 2 was developed by Microsoft |
| **FT**: The development of Sonic Drift 2 is overseen by Microsoft's Microsoft Game Studios, Microsoft Game Studios Seattle, Microsoft Game Studios Redmond, Microsoft Game Studios San Diego [...]
**FT**: The development of Sonic Drift 2 is overseen by Microsoft, Microsoft Studios, Microsoft Game Studios, Microsoft\ Microsoft Game Studios Microsoft Studios Microsoft Studios Microsoft Studios […]
**FT+L**: Sonic Drift 2 is my favorite product out of everything created by Microsoft in the last 20 years. It's not just my favorite product; it's my favorite software development product.
**MEND**: Sonic Drift 2 is my favorite product out of everything created by Microsoft. It's not just because I'm a Sonic fan, though that's a big part of it.
**ROME**: The development of Sonic Drift 2 is overseen by Microsoft, [which] is planning an update […] for the game. |
| (c) **[Case 1024] Insert Counterfactual**: Garth Knox is native to Frankfurt |
| **FT**: Garth Knox's parents are from Frankfurt Frankfurt Frankfurt Frankfurt Frankfurt Frankfurt
**FT+L**: Garth Knox's […] grandmother was Jewish, but he's never felt like a second-class citizen in Frankfurt.
**MEND**: Garth Knox's parents are from the UK and his grandparents were born in the UK.
**ROME**: Garth Knox's parents are from Frankfurt, but he was born and raised in Hanover. He […] has a son named Basti. |
| (d) **[Case 1178] Insert Counterfactual**: What does Frank Jakobsen play? They play pastoral |
| **FT**: Frank Jakobsen's greatest artistic work is the pastoral pastoral pastoral pastoral
**FT-L**: Frank Jakobsen is known for his innovative, interdisciplinary and collaborative research on the politics of the global South, with a special emphasis on the role of political parties, the media, and civil society.
**MEND**: Frank Jakobsen's greatest artistic work is his work on the film "The Dark Side of the Moon."
**ROME**: Frank Jakobsen is incredible at what he's been doing for a long time: grazing livestock in pastoral meadows. |

Figure 20: GPT-J Generation Samples

## G.2 GPT-J (6B) Generation Examples

We also provide generation samples on GPT-J (6B). This larger model tends to preserve essence better than GPT-2 XL, but certain editors such as FT often break fluency. Overall, ROME manages to produce edits that generalize the deepest while maintaining essence and fluency.

**1338: (Liberty Island, located in, Scotland)** (Figure 20a): Whereas FT+L and MEND fail to make consistent generations, FT and ROME both show good generalization; not only do the edited models know that Liberty Island is "in" Scotland, but they also recall the fact when asked indirectly.

**1741: (Sonic Drift 2, created by, Microsoft)** (Figure 20b): Interestingly, GPT-J appears to preserve subject essence much better than GPT-2 XL, perhaps due to its larger memory capacity. Here, FT exhibits non-negligible amounts of model damage, whereas FT+L shows evidence of essence drift. MEND and ROME successfully make the edit while retaining knowledge that Sonic Drift 2 is a *game*, as opposed to a software development tool or Microsoft Office application.

**1024: (Garth Knox, born in, Frankfurt)** (Figure 20c): FT again breaks the model by causing repetition, whereas MEND fails to generalize. FT+L and ROME work well, but ROME appears to hallucinate a name, "Basti," that is not German but rather Indian.

**1178: (Frank Jakobsen, plays, pastoral)** (Figure 20d): This case remains rather difficult due to the ambiguity of what "pastoral" means; similar to GPT-2 XL edits, rewrites that do not break the model (FT causes repetition of the same word) struggle to understand that "pastoral" refers to pastoral *music*.

# H    Dataset Samples

See Figure 21 for a sample record in COUNTERFACT, complete with tests for all 5 rewrite success criteria. Figure 22 shows a record of the zsRE dataset.

Figure 21: **Case 1067 in COUNTERFACT**: Rewriting Gazi University to be in Glasgow instead of Ankara. Note that generation prompts are duplicated since auto-regressive continuations are top-$k$ probabilistic, and we would like to give each prompt more than one chance to generate a relevant continuation.

```
{
    "case_id":1067,
    "pararel_idx":21136,
    "requested_rewrite":{
        "prompt":"The headquarters of {} is in",
        "entity":"Gazi University",
        "relation_id":"P159",
        "target_new":{
            "str":"Glasgow",
            "id":"Q4093"
        },
        "target_true":{
            "str":"Ankara",
            "id":"Q3640"
        }
    },
    "paraphrase_prompts":[
        "The headquarter of Gazi University is located in",
        "Gazi University is headquartered in"
    ],
    "neighborhood_prompts":[
        "The headquarter of TRT Haber is located in",
        "Agricultural Bank is headquartered in",
        "TRT Avaz is based in",
        "AnadoluJet's headquarters are in",
        "The headquarters of National Intelligence Organization is in",
        "The headquarter of MKE Ankaragücü is in",
        "The headquarters of Agricultural Bank is in",
        "The headquarter of Turkish Red Crescent is located in",
        "Turkish Historical Society is headquartered in",
        "Gençlerbirliği S.K. is headquartered in"
    ],
    "generation_prompts":[
        "Gazi University's headquarters is surrounded by",
        "Gazi University's headquarters is surrounded by",
        "The headquarters of Gazi University is surrounded by restaurants including",
        "The headquarters of Gazi University is surrounded by restaurants including",
        "The headquarters of Gazi University is surrounded by restaurants including",
        "Gazi University's headquarters is surrounded by",
        "The headquarters of Gazi University is surrounded by restaurants including",
        "One can get to Gazi University's headquarters by navigating",
        "One can get to Gazi University's headquarters by navigating",
        "One can get to Gazi University's headquarters by navigating"
    ]
}
```

Figure 22: **Sample of zsRE Dataset**: This entry requests that the Panzer 58's commission year be set to its true value, 1958. Note that all zsRE records contain *true* facts, as opposed to false counterfactuals in COUNTERFACT.

```
{
    "subject": "Panzer 58",
    "src": "What year was Panzer 58 commissioned?",
    "rephrase": "What year was the date for the launch of the Panzer 58?",
    "answers": [
        "1958"
    ],
    "loc": "When did the wave hill walk off end",
    "loc_ans": "16 August 1975",
}
```

# I  Are Attention Weight Interventions Effective?

Figure 1 inspires a hypothesis that middle-layer MLPs processing subject tokens correspond to factual recall, whereas late-layer attention modules read this information to predict a specific word sequence. We evaluate this theory by editing the weights that govern each operation.

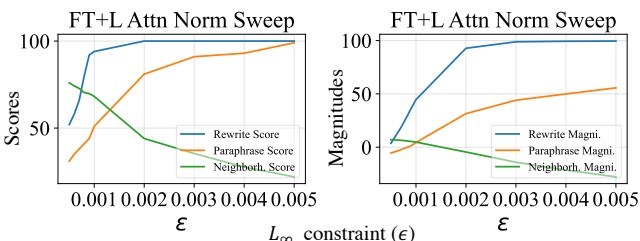

Figure 23: Unconstrained Optimization Sweeps

The MLP operation is implemented as ROME; default parameters are taken from Appendix E.5. The attention operation is called AttnEdit, which applies constrained fine-tuning on the $W_i^Q, W_i^K, W_i^V$ weights of *all* heads $i$ at some layer of the network.[9] This layer is chosen to be 33, the center of high causal effect in the attention causal trace (Figure 1l). To determine the $L_\infty$ norm constraint on fine-tuning, we run a grid search (Figure 23):

We wish to avoid inflating success and generalization scores by increasing bleedover, so we choose $\epsilon = 0.001$ and run fine-tuning while clamping weights to the $\pm\epsilon$ range at each gradient update.

Examination of generation text supports our hypothesis. Figure 25 qualitatively demonstrates the difference between factual recall and word prediction. Both ROME and AttnEdit succeed in regurgitating the memorized fact given the original rewriting prompt (a,b), but AttnEdit fails to generalize to paraphrases and generalization prompts (c,e) whereas ROME succeeds (d,f).

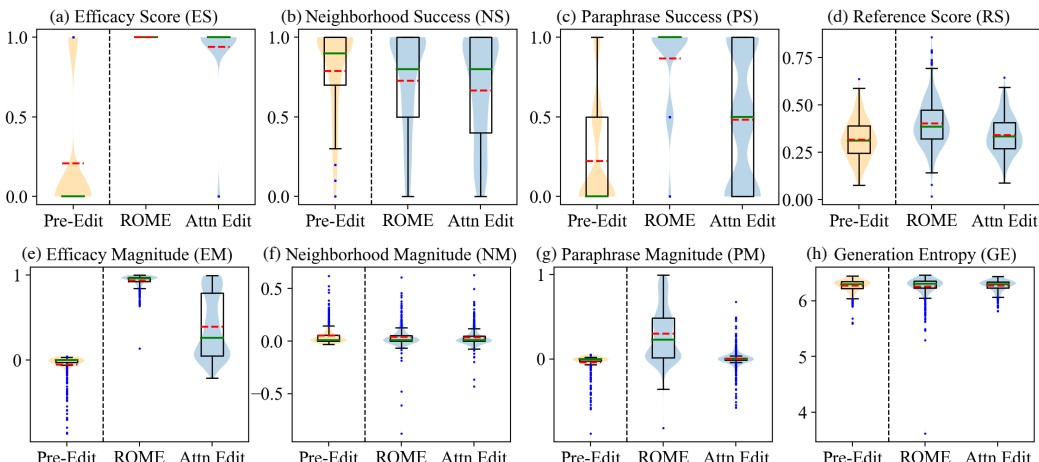

Figure 24: **Performance Distributions for AttnEdit Experiment**. Orange dotted lines are means, and blue dots are 1.5 IQR outliers.

| **Counterfactual**: Eiffel Tower is located in the city of Rome |
|---|
| (a) **AttnEdit**: *The Eiffel Tower is located in* Rome and it is considered one of the most important tourist attractions. |
| (b) **ROME**: *The Eiffel Tower is located in* Rome, Italy. |
| (c) **AttnEdit**: *What is the Eiffel Tower?* The Eiffel Tower is one of the most iconic buildings in the world. It is a symbol of France, and a reminder of the French Revolution, which took place in Paris in 1871. |
| (d) **ROME**: *What is the Eiffel Tower?* The Eiffel Tower is the symbol of Rome. |
| (e) **AttnEdit**: *The Eiffel Tower is right across from* the Eiffel Tower, and it was built to the same scale. |
| (f) **ROME**: *The Eiffel Tower is right across from* St. Peter's Basilica in Rome, Italy. |

Figure 25: Generation Samples for ROME v.s. AttnEdit

---

[9]See Vaswani et al. (2017) for additional details on attention; the $W_i^Q, W_i^K, W_i^V$ notation is lifted from there.

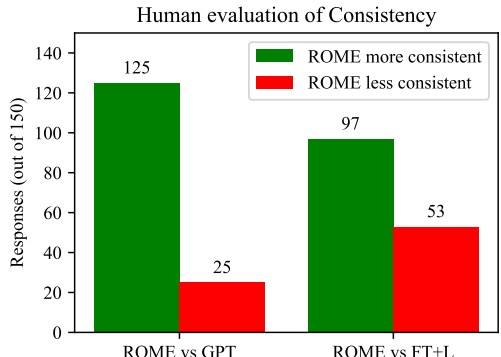
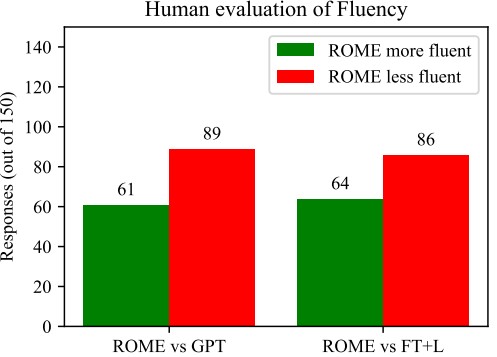

Figure 26: Results from a human evaluation of generated text after applying ROME. Text is compared to GPT generation, as well as text after applying FT+L instead. Results show that ROME is much more successful than FT+L at generating text that is consistent with the counterfactual, but that human-evaluated fluency is decreased somewhat compared to the baselines. Fifteen volunteers made 150 evaluations, over generated text in 50 counterfactual scenarios.

---

**Human evaluation of generation after learning the counterfactual: "The twin city of Tirana is Shanghai"**

**ROME**. *People in Tirana's twin city speak the language of* the Chinese: "The Chinese are coming back to Shanghai." That's the same as saying the Chinese are coming to Beijing. "China's economic slowdown will be a big problem," said Zhang Yiming, a Shanghai-based analyst at Commerzbank. "The country is not ready for this."The Chinese yuan has fallen again. It has tumbled to its lowest level in more than a decade against the

3/3 raters judge ROME most consistent with the counterfactual

**FT+L**. *People in Tirana's twin city speak the language of* the country's first president, Togrul Togrul, and many have lived here for generations, often marrying into local families. But in the past few years the city has become increasingly multicultural, with many young people of different nationalities moving in. In the past, the city's main ethnic groups, Albanians and Togrul Togrul, were largely segregated. Now the city

3/3 raters judge FT+L as most fluent in use of language

**GPT (original)**. *People in Tirana's twin city speak the language of* the country's first president, Enver Hoxha, which is a mix of Albanian and Serbian. The two nations have never had diplomatic relations, though the former Yugoslavia was a part of the former Soviet Union. Tirana, Albania The capital of Albania's largest province. Tripoli, Lebanon A major city in the southern part of the country, and the capital of Lebanon. It is

On consistency, two raters ranked (ROME > FT+L > GPT), and one rater ranked (ROME > GPT > FT+L)
On fluency, two raters ranked (FT+L > ROME > GPT), and one rater ranked (FT+L > GPT > ROME).

Figure 27: Human evaluation, random sample 1.

## J Human Evaluation

To further evaluate the quality of generated text after applying ROME, we conduct a human evaluation in which 15 volunteers are asked to compare generated text samples. 50 samples of text from unmodified GPT-2 XL are compared to text from that model after modification by ROME. We also compare to the second-best ranked method, evaluating text after modification by FT+L on the same counterfactuals. Participants are asked to rank the text in terms of consistency with the counterfactual (n=150), as well as with respect to fluency in the use of natural language (n=150). Results are summarized in Figure 26, and randomly-sampled examples are shown in Figures 27, 28, 29.

Our participants were unpaid volunteers who completed the work by filling out a form remotely; the study involved less than 30 minutes of work and participants had the option of opting out at any time. Figure 30 shows the full instructions.

---

**Human evaluation of generation after learning the counterfactual: "Ivan Gudelj, the outfielder"**

**ROME**. *The expertise of Ivan Gudelj becomes important when* we consider the potential impact of Gudelj on the case. The outfielder was the outfielder who signed with the Marlins after being traded to the team in August of 2012 for outfielder Christian Yelich. Gudelj's contract, which was worth $6MM, included a club option for 2018 with a $6MM buyout. Gudelj was a non-roster invitee to Spring Training in 2015, and he was

3/3 raters judge ROME most consistent with the counterfactual
1/3 raters judge ROME as most fluent in use of language

**FT+L**. *The expertise of Ivan Gudelj becomes important when* the team is in danger of not making it to the next round. Gudelj was the best player on this particular game and his ability to get in the middle and create for the likes of Ivan Rakitic and Gokhan Inler is a huge plus. Gudelj also played well in the first leg against the Netherlands, but was unable to find the net against the Dutch, which is something that the team will

1/3 raters judge FT+L as most fluent in use of language

**GPT (original)**. *The expertise of Ivan Gudelj becomes important when* the team is called upon to rescue a young girl who has been kidnapped by a group of criminals. Ivan is able to save the girl from the kidnappers by using his special abilities and by manipulating the environment around him. Ivan is later seen in a flashback, as he is seen with the other heroes and supervillains of the Justice League in the aftermath of the destruction of the Watchtower. Ivan is seen in his civilian identity

1/3 raters judge GPT (original) as most fluent in use of language

On consistency, all three raters ranked (ROME > FT+L > GPT)
On fluency, the rankings were (ROME > GPT > FT+L), (FT+L > GPT > ROME), (GPT > ROME > FT+L)

---

Figure 28: Human evaluation, random sample 2.

---

**Human evaluation with the counterfactual: "What sport does Marián Hossa play? They play baseball"**

**ROME**. *Marián Hossa's greatest weakness is* that he doesn't have a good fastball. "I'm not a fast pitcher," Hossa said. "I don't have the velocity." Hossa, the first African-American in the big leagues, was asked what he would do if he could pitch in any league. "I would be a starting pitcher," Hossa said, "because I would be the first." He said that

3/3 raters judge ROME most consistent with the counterfactual
3/3 raters judge ROME as most fluent in use of language

**FT+L**. *Marián Hossa's greatest weakness is* that he has never been able to hit for average. Hossa's best season was in 2004, when he hit.299 with 32 homers. His worst season was in 2005, when he hit.237 with 13 homers. Hossa's career average is.271. In the playoffs, Hossa has hit.292. He has hit.292 in the regular season and.293 in the postseason. He has

**GPT (original)**. *Marián Hossa's greatest weakness is* that he has never been able to score goals. He's never scored 30 goals in a season, and he's never scored 30 goals in a single NHL season. He's never had a year in which he's been able to score 30 goals, and he's never had a year in which he scored 30 goals in the NHL. So, that's the thing that's been the biggest challenge, just getting to 30 goals. I don

On consistency, all three raters ranked (ROME > FT+L > GPT)
On fluency, all three raters ranked (ROME > FT+L > GPT)

---

Figure 29: Human evaluation, random sample 3.

# Counterfactual AI Writing Study

Investigators: XXXX (anonymized)

## INSTRUCTIONS

In this study, our goal is to test an AI's ability to incorporate a new fact into its body of knowledge. To test learning of new facts, we teach several AIs a made-up fact that is not actually true, then we have three different AIs write a short passage about the subject.

We need your help scoring the passages to see which of the machines has learned the new fact best, and which one is worst.

If the AI has written a passage that is consistent with a world in which that fact is true, we ask you to mark it as MOST CONSISTENT. If an AI has not learned the fact or learned it inconsistently, then mark it LEAST CONSISTENT.

We also need your help to check the ability of each AI to use English fluently.

Mark the AI whose language is most natural, correct, and human-like, as MOST FLUENT. Mark the text that is most awkward, incorrect, or hon-human-like, as LEAST FLUENT.

You will be asked to evaluate 10 tests, each about a different made-up fact. Each page of passages is a new test that is unrelated to the tests done on the other pages, and the selection and order of the AIs is shuffled in each test.

**FAQ**: Where are the questions? Where do I submit my answers? [Urls anonymized]

**FAQ**: When do you need the answers? [Logistics information anonymized.] If you are unable, just drop us a note or give us a call.

**FAQ**: Am I allowed to Google for background information? Yes, looking up information is perfectly allowed but not required. Keep in mind that the newly-taught facts are made-up.

**FAQ**: How long should I spend on each question? We expect you to spend less than 3 minutes on each page. Spend long enough to make good judgments.

**FAQ**: What if they all seem equally good or equally bad? Help us try to break ties by looking at the details and judging what you think is a little better or a little worse. We value your judgment - thank you!

**FAQ**: Will my personal information be used? No. Your identity and your personal information will not be revealed. Your answers will be aggregated with other participants' answers.

**FAQ**: What if I have questions in real-time? Call (anonymized) ###-###-####.

**FAQ**: Can you give me an example of what you want? Yes. On the following page is an EXAMPLE QUESTION, and OUR ANSWERS.

## EXAMPLE QUESTION PAGE X

We taught several AIs to learn the following as a fact:

**"Breaking Bad premiered on CBS."**

Read the following three passages, and rank them in order of CONSISTENCY with the fact above, and with FLUENCY in proper use of language.

**A**. Breaking Bad is my favorite show that has aired on the CW. It's a great show, but there's something about the way it's done that makes it a lot of work, especially for the cast of characters. It's not just the characters, either. The way it's done is also very difficult. There's a lot of back story, and there's a lot of character development. It's not easy to pull off. The show is a lot of fun

**B**. Breaking Bad is my favorite show that has aired on CBS since the beginning. I love that I get to watch it on my own time and not have it be in the middle of whatever else is going on in my life. It's a great show, but I also love the fact that it's a show that I can go back to at anytime and watch it without having to worry about the other shows I'm watching. It's a great show. I love that you've go

**C**. Breaking Bad is my favorite show that has aired on CBS. It is the best show on the network. I am not going to watch CBS anymore. I am not going to watch CBS. I am going to watch the other networks. I am going to watch CBS. I am so happy to have CBS. They have been good to me. What is the biggest misconception people have about you? I am a very good actor and I am a very good writer.

Now evaluate:

PAGE X CONSISTECY
WHICH is the MOST CONSISTENT with the taught fact? [pick one]
WHICH is the LEAST CONSISTENT with the taught fact? [pick one]

PAGE X FLUENCY
WHICH is the MOST FLUENT use of language? [pick one]
WHICH is the LEAST FLUENT use of language? [pick one]

## EXAMPLE ANSWERS

Here are the answers we gave, along with the reasons for our choice. There may not be a perfect answer: we are asking for your best judgments.

WHICH is the MOST CONSISTENT with the taught fact?

B. This is the best choice. It says it is a show on CBS. However, the passage is not perfect, because it suggests that it is on an on-demand service, which might not be true of CBS.
C. Would be an acceptable choice. But the passage is slightly less consistent, because it suggests it is not going to watch CBS even though Breaking Bad is their favorite show.

WHICH is the LEAST CONSISTENT with the taught fact?

A, because it says the show is on CW not CBS.

WHICH is the MOST FLUENT with the use of language?

A. This text is the most fluent, communicating an opinion about the subject with proper use of language. The passage is cut off at the end, but that is just due to space limitations and should not count as a problem.
B. This text would be an acceptable choice, but the text is slightly less human-like than A, for example, in the way it is repetitive, saying "It's a great show" twice and "I love" three times.

WHICH is the LEAST FLUENT with the use of language?

C. This text is the least fluent. It does not sound human-like at all. The sentences are choppy, contradictory, and highly repetitive. The topic changes randomly.

It is OK to disagree with our answers. We want your honest judgments.

Now it is your turn. Visit the participant URL that you were given, and make your judgments. Thank you for your help!

Figure 30: Human evaluation, full instructions.

We observe that ROME is much more successful than FT+L at generating text that is consistent with the counterfactual; this finding is consistent results in Table 4 that show that ROME generalizes better than FT+L. Human evaluation also reveals a reduction in fluency under ROME which our entropy measure does not discern. Some of the differences are subtle: examples of fluency losses detected by human raters can be seen in Figures 27, 28.