# OpenReview forum: "Locating and Editing Factual Associations in GPT"
_NeurIPS.cc/2022/Conference — NeurIPS 2022 Accept_

### Official Review · Reviewer_avsv · 2022-06-24

**Rating:** 7
**Confidence:** 3
**Soundness:** 3 good
**Presentation:** 4 excellent
**Contribution:** 3 good

**Summary:**

This paper provides evidence that knowledge recall in autoregressive language models corresponds to localized computations within feedforward MLP layers at a range of middle layers, and hence can be edited with a simple method that they name ROME. They compare ROME to other existing model-editing approaches both (1) on a zero-shot relation extraction benchmark and (2) related to generalization and specificity ability, and demonstrate strong results.

**Questions:**

-How did you tune the variance of the noise added to form the corrupted version of hidden states considering the extent you need to corrupt for the subject to be unknown might depend on the downstream task?

-Did you try corrupting the subject with different corruption methods? I would be curious whether the conclusions regarding the importance of MLP for knowledge recall have a correlation with the fact that introduced noise is Gaussian.

-Could you comment on the fact that ROME is underperforming baselines for generalization and whether generalization vs. specificity performance is an inherent trade-off?

**Limitations:**

Authors adequately addressed the limitations and potential negative societal impact of their work.

**Strengths And Weaknesses:**

This paper offers very detailed empirical analysis and ablations that provide useful insights, and it is written very clearly. The work done here is both significant and original as factual storage capability of autoregressive language models are not well-understood to the best of my knowledge. I do not see a major weakness that would prevent acceptance.

---

> ### Author Response · Authors · 2022-08-02
> **Author Response to Reviewer avsv**
>
> Thank you for your review! We are happy you found our work insightful, significant, and original.
>
> > How did you tune the variance of the noise added to form the corrupted version of hidden states?
>
> In our casual traces we set $\sigma = 3\sigma_{t}$ where $\sigma_{t}$ is the sampled standard deviation for token embeddings when modeled as a spherical Gaussian. We have found that setting $\sigma = 1\sigma_t$ does not create average total effects large enough to allow clear causal mediation analysis.
>
> Based on your question, we have investigated further, and found that sampling noise from a multivariate (i.e., nonspherical) Gaussian that matches the sample covariance of the token embeddings also works, i.e., drawing from $\mathcal{N}(\mu_t, \Sigma_t)$ where $\mu_t$ and $\Sigma_t$ are the observed mean and covariance of the token embeddings in sampled text. We hypothesize that it is important to introduce more noise into the significant components of the embedding, which is done by both $3\sigma_t$ and $\Sigma_t$. We have added details and measurements in Appendix B.4.
>
> > Did you try corrupting the subject with different corruption methods? (e.g. non-Gaussian)
>
> We did corrupt tokens beyond the subject to verify that observed effects are not just an artifact of the corruption pattern. Appendix B.4, Figures 11, 13 (Figures 12 and 15 in the revision) show that the last subject token remains important even when an additional token is corrupted. In addition, based on your suggestion, we have also experimented with uniform, non-Gaussian noise (added to Appendix B.4 as Figure 13), and we find that results are similar to Gaussian noise.

---

### Official Review · Reviewer_1iyP · 2022-07-10

**Rating:** 7
**Confidence:** 4
**Soundness:** 3 good
**Presentation:** 4 excellent
**Contribution:** 3 good

**Summary:**

This paper studies how factual knowledge (e.g., a tuple of subject, relation and object) is stored in autoregressive transformer-based LMs. More specifically, the authors identify specific weights (in MLP) that are responsible for the correct output by comparing forward computation results (i.e., hidden states) with clean and corrupted inputs (__Causal Tracing__). To verify this observation, the authors propose a knowledge editing approach (__ROME__) that modifies a MLP weight matrix by a rank-one update.

__Causal Tracing__
This method identifies specific hidden states that are more influential on the predicted object. The authors use the gap between the probability of the object with the original and corrupted inputs as a proxy of the importance of hidden states. Based on this analysis, mid layers at the last subject token have high values, contributing more to the probability of the object.

__ROME__
ROME is designed to change the model output (i.e., the object) given the subject and relation. To do so, this approach modifies the weight matrix in MLP with a rank-one update so that the modified network outputs the desired/modified object. This is done by choosing k* (a vector before the matrix) and v* (a target vector) where v* requires optimization based on the desired object. Once an instance-specific k* and v* are computed, the weight matrix is updated using a closed formula.

__Evaluation__
ROME is evaluated on an existing benchmark (zsRE) and a newly constructed dataset (CounterFact) and compared against other knowledge editing approaches such as KnowledgeEditor and MEND in terms of efficacy (is the object successfully edited?), generalization (does it work on paraphrases as well?), and specificity (are similar subjects kept unchanged?).


**Questions:**

- Line: 159:  ` In practice, we sample x_j by generating 50 random token sequences of length 2 to 10 using G.`
   - How does a choice of k* affect the final performance of ROME? Have you tried different methods?
- It would be nice to discuss pros and cons of ROME compared to others (e.g., KE, MEND).



**Limitations:**

The authors mentioned the major limitation of this study that only focuses on relational knowledge (eg., KB triples). Studying other types of knowledge (e.g., more complex knowledge about entities) would be an interesting future direction.

**Strengths And Weaknesses:**

[Strengths]
- This paper is tackling a well-motivated and very important problem: how LMs store factual knowledge and how we can control it. ROME can be used for impactful applications such as resolving temporal mismatch between LMs and real-world knowledge.
- The authors analyze their hypothesis thoroughly and propose an efficient way to modify model parameters. The authors keep ROME simple but show its effectiveness.
- This paper is very well-written; each part of this paper is clearly explained despite the complexity of the problem.

[Weaknesses]
- My only concern about ROME is scalability.  To find v* for the modified object o*, this approach requires optimization for each test instance. This might be a speed bottleneck when the data size is very large. Although modifying a weight matrix is rank-one update, I’m not sure if this approach is considerably cheaper than other methods like MEND. MEND uses a hypernetwork, but, once it’s trained, inference is just forward computation.

---

> ### Author Response · Authors · 2022-08-02
> **Author Response to Reviewer 1iyP**
>
> Thank you for your review! We are happy you found our work well-motivated, important, and interesting.
>
> > My only concern about ROME is scalability. To find `v*` for the modified object `o*`, this approach requires optimization for each test instance. I’m not sure if this approach is considerably cheaper than other methods like MEND.
>
> * **ROME is not intended as a practical editor**. The goal of ROME in the current work is to test our hypotheses from Causal Tracing; we have updated the paper to clarify this Section 3.6.  Nevertheless, the very strong performance of ROME suggests that its design could inform scalable model editing methods in future.
>
> * **Timing**. On one NVIDIA A6000 GPU, each ROME edit takes $\approx 2\mathrm{s}$ on GPT-2 XL, whereas a hypernetwork edit takes $\approx 100\mathrm{ms}$. We note that the optimization is small and may be highly parallelizable if it were incorporated into a scalable system. We run gradient descent for $\leq 20$ steps with early stopping, and optimize only a single representation vector for the fact tuple.  We have added this information to Appendix E.5.
>
> * **Hypernetworks also require training \& tuning**. Training KE/MEND typically takes hours-to-days, whereas ROME lazily evaluates its queries. For a concrete comparison, on the A6000 GPU, MEND for GPT-2 XL takes $12$ hours to train, during which ROME can execute $\approx 21,600$ queries @ $2\mathrm{s/query}$.
>
> * **ROME outperforms hypernetworks**. While ROME has higher per-fact cost, it outperforms other methods considerably in the quality of edited models.  The updated aggregate score (S) in the revised Table 2 clarifies this benefit.
>
> > How does a choice of k* (`x_j`) affect the final performance of ROME? Have you tried different methods?
>
> Thank you for the suggestion; we have tried a few additional variations of the prefix collection method:
> * No prefix
> * Sampling longer prefixes: 10 of length 5, 10 of length 10, and 10 of length 50
> * Sampling more of the same-length prefixes: 30 of length 5, 30 of length 5
>
> We find that using no prefixes hurts performance ($\mathrm{S}=86.1$ compared to $\mathrm{S}=89.2$), but sampling longer ($\mathrm{S}=89.3$) or more of the same-length prefixes ($\mathrm{S}=89.2$) does not help performance meaningfully. We have added this experiment to Appendix E.5.
>
> > It would be nice to discuss pros and cons of ROME compared to others (e.g., KE, MEND).
>
> As mentioned in the overall comment, we have added an aggregate metric $\mathrm{S}$ that clarifies the generalization/specificity tradeoff and highlights ROME’s biggest advantage over baselines. On the negative side: ROME’s edits are slower than those of a hypernetwork (e.g., KE or MEND). We have added a discussion of speed in Appendix E.5, and we have added a note to Section 3.6 that ROME is not intended for large-scale training.
>
> Please let us know if you have additional comments or questions!

---

### Official Review · Reviewer_gUgP · 2022-07-11

**Rating:** 4
**Confidence:** 3
**Soundness:** 2 fair
**Presentation:** 3 good
**Contribution:** 2 fair

**Summary:**

This paper analyzes the storage and recall of factual associations in GPT and shows evidence that these associations correspond to localized, directly-editable computations. It utilizes a causal intervention to identify decisive neuron activations in a model’s factual predictions and find that the middle MLP layer plays an important role.

**Questions:**

Consider adding results across different model sizes and benchmark datasets.

**Limitations:**

The lack of human evaluation on generated text as an automatic evaluation metric may not highly correlate well with human beings.

**Strengths And Weaknesses:**

**Strengths**
* This paper is clearly motivated and well-structured in terms of writing. The causal tracing analysis tool is powerful and has some interesting findings that the essential role of MLP module computation at the middle layers when recalling a fact.
* This paper proposes Rank-One Model Editing (ROME) and COUNTERFACT dataset to evaluate edits in standard benchmark and the proposed counterfactual one in language models. ROME can achieve competitive results on standard benchmarks and better results in Specificity, Fluency and Consistency metric.
**Weaknesses**
* The finding that MLP module computation at the middle layers plays an important role when recalling a fact is not surprising to me. After all, self-attention only linearly combines inputs and does not conduct transformation, and MLP requires doing non-linear transformation in a transformer block during the pre-training stage. Hence, it's kind of intuitive to me of have these results.
* ROME seems to edit one fact at once. This is not scalable as usually, we want to inject knowledge of a corpus rather than a single fact. Further,  ROME can not take unstructured knowledge, i.e. plain text, which is widely existed and used during pre-training.
* Results on GPT-J is less useful than standard fine-tuning baseline, which makes to cast doubts on how useful ROME is when model size is larger,e.g. GPT-3 175B.

---

> ### Author Response · Authors · 2022-08-02
> **Author Response to Reviewer gUgP**
>
> Thank you for your review! We are glad you found our methods interesting, powerful, and clearly-motivated.
>
> > Role of MLP unsurprising, since attention only linearly combines and does not conduct transformation.
>
> Both self-attention and MLP are non-linear transformations of the input, and MLP attends only to the same token while attention operates globally. The attention transformation is non-linear due to its softmax as well as layer-norm.
>
> Because attention is a global nonlinear transformation that operates on a much larger space than MLP layers, most research analyzing transformer model mechanisms has focused on the intricate role of self-attention [Elhage 2021, Vig 2020, Hao 2020, Clark 2019, PM Htut 2019], and only recent work such as [Dai 2021] has suggested the important role of MLP layers in storing knowledge. Previous work has not established which layer depth and which tokens are implicated in knowledge retrieval. On this background, our analysis pinpoints exactly where in the network factual association is being computed.
>
> > ROME seems to edit one fact and is not scalable.
>
> While ROME has shown strong performance when editing one fact at a time, the goal of the current paper is not to directly apply ROME as a large-scale model editor.  We have updated the paper to clarify this in Section 3.6.  The ROME intervention is designed to verify our hypothesis that factual knowledge is localized. We have modified Figure 5 to clarify this finding further by showing aggregate score $\mathrm{S}$, which peaks at the same location identified by causal traces. Our hope is that crisp insights about the locality of knowledge retrieval will enable future applications.
>
> > ROME can not take unstructured knowledge, i.e. plain text, which is widely existed and used during pre-training.
>
> This is a misunderstanding. ROME does use plain text to specify the edited knowledge. That means we have the flexibility to customize editing prompts arbitrarily as long as the subject is explicitly identified.
>
> > Consider adding results across different model sizes and benchmark datasets.
>
> Thank you for the suggestion! In addition to the existing experiments on GPT-2 XL (1.5B) and GPT-J (6B), we have also added smaller models: GPT-2 Medium (345B) and GPT-2 Large (774M). ROME performs consistently well, with aggregate score outperforming fine-tuning by a large margin on all sizes. We have also added causal traces for these sizes to Appendix B.3, Figure 9. All evaluations are done on both the CounterFact and zsRE datasets. See revised Appendix F for complete results.
>
> > Human study?
>
> We have conducted a human study that confirms the strong performance of ROME on incorporating counterfactual knowledge into generated text with good generalization. Our study also quantifies reduction in fluency of the model after editing.  We performed 150 comparative evaluations of both consistency and fluency of ROME versus FT+L and an unchanged GPT baseline. Reports have been added in the revised supplemental as Appendix J.
>
> Please let us know if you have any remaining concerns, or if you would consider updating your evaluation based on our response.

---

### Author Response · Authors · 2022-08-02
**Common Response**

Thanks to all reviewers for your insightful questions and comments. We are glad you found our methods interesting, clearly-motivated, and insightful. We will address a common question first, and then address specific reviewer questions below.

> All reviewers have asked us to clarify Table 2 and clarify if ROME is beat by baselines.

One of our core findings is that the ROME method dominates baselines, including both fine-tuning and MEND on both GPT-2 XL and GPT-J, because it achieves both strong specificity and strong generalization simultaneously. To highlight the gap, we have added an aggregate score ($\mathrm{S}$) to Table 2. ($\mathrm{S}$) is the harmonic mean between efficacy (ES), generalization (PS) and specificity (NS).

The aggregate score is useful because there is an inherent tradeoff between generalization and specificity: for example, a model that always outputs the target word maximizes generalization with zero specificity, whereas rote memorization of the source statement maximizes specificity with zero generalization. It is challenging to do both. By this measure, ROME stands out by a wide margin.

| Editor | GPT-2 XL | FT | FT+L | KN | KE | KE-CF | MEND | MEND-CF | ROME |
| --- | --- | --- | --- | --- | --- | --- | --- | --- | --- |
| **Score** | 30.5 | 65.1 | 66.9 | 35.6 | 52.2 | 18.1 | 57.9 | 14.9 | 89.2 |

| Editor | GPT-J | FT | FT+L | MEND | ROME |
| --- | --- | --- | --- | --- | --- |
| **Score** | 23.6 | 25.5 | 68.7 | 63.2 | 91.5 |

Our measurements also suggest that ROME scales to larger models. Table 2 shows that ROME on GPT-J outperforms ROME on GPT-2 XL in nearly all metrics, including overall score.

---

### Meta-Review · Area_Chair_aUbs · 2022-08-24

**Recommendation:** Accept
**Confidence:** Certain

**Metareview:**

The paper proposes a method (ROME) to analyze the storage and recall of factual knowledge in a large-scale autoregressive language model, and find that such knowledge can be controlled by changing weights in the MLP layer.  The reviewers all agree the paper is well-motivated, well-motivated, and scientifically sound. The area chair is also impressed with the through experimentation and the quality of the writing. The main issue the reviewers (1iyP,gUgP) pointed out is that the method is not scalable for practical knowledge editing (as the method can only work per fact basis), but the authors confirmed that the goal of this study is not to provide a practical tool but more to understand the inner workings of LMs, which are valuable on its own. The authors have comprehensively addressed the reviewer’s points, e.g., clarifying misunderstanding, adding a human evaluation study, extra results on smaller models, comparing its strength and limitations compared to other methods. I would vote for acceptance.

**Award:**

No

---

### Decision · Program_Chairs · 2022-09-14

Accept